

# Inversion of multi-angular polarimetric measurements over open and coastal ocean waters: a joint retrieval algorithm for aerosol and water leaving radiance properties

Meng Gao[1,2], Peng-Wang Zhai[1], Bryan Franz[3], Yongxiang Hu[4], Kirk Knobelspiesse[3], P. Jeremy Werdell[3], Amir Ibrahim[2], Brian Cairns[5], and Alison Chase[6]

[1]JCET, Physics Department, University of Maryland, Baltimore County, Baltimore, MD 21250, USA
[2]SSAI,NASA Goddard Space Flight Center, Code 616, Greenbelt, Maryland 20771, USA
[3]NASA Goddard Space Flight Center, Code 616, Greenbelt, Maryland 20771, USA
[4]MS 475 NASA Langley Research Center, Hampton, VA 23681-2199, USA
[5]NASA Goddard Institute for Space Studies, New York, NY 10025, USA
[6]School of Marine Sciences, University of Maine, Orono, ME 04469, USA

**Correspondence:** Peng-Wang Zhai (pwzhai@umbc.edu)

**Abstract.** Ocean color remote sensing is a challenging task over coastal waters due to the complex optical properties of aerosols and hydrosols. In order to conduct accurate atmospheric correction, we previously implemented a joint retrieval algorithm to obtain the aerosol and water leaving signal simultaneously. The algorithm has been validated with synthetic data generated by a vector radiative transfer model and good retrieval performance has been demonstrated in terms of both aerosol and

5     ocean water optical properties [Gao et al., Optics Express **26**, 8968–8989 (2018)]. In this work we applied the algorithm to airborne polarimetric measurements from the Research Scanning Polarimeter (RSP) over both open and coastal ocean waters acquired in two field campaigns: the Ship-Aircraft Bio-Optical Research (SABOR) in 2014 and the North Atlantic Aerosols and Marine Ecosystems Study (NAAMES) in 2015 and 2016. Two different yet related bio-optical models are designed for ocean water properties. One model aligns with traditional open ocean water bio-optical models that parameterize the ocean

10    optical properties in terms of the concentration of chlorophyll a. The other is a generalized bio-optical model for coastal waters that includes seven free parameters to describe the absorption and scattering by phytoplankton, colored dissolved organic matter and non-algal particles. The retrieval errors of both aerosol optical depth and the water leaving radiance are evaluated. Through the comparisons with ocean color data products from both in situ measurements and the Moderate Resolution Imaging Spectroradiometer (MODIS), and the aerosol product from both the High Spectral Resolution Lidar (HSRL) and the Aerosol

15    Robotic Network (AERONET), our algorithm demonstrates both flexibility and accuracy in retrieving aerosol and water leaving radiance properties under various aerosol and ocean water conditions.



# 1 Introduction

The ocean is of immense importance for Earth's climate and ecosystems, and its conditions have great economic and social impacts (Costanza, 1999). It is critical to monitor and evaluate oceanic biogeochemical properties on the global scales using approaches such as ocean color remote sensing (Chapman, 1996). For both spaceborne and airborne remote sensing of ocean

color, atmospheric correction is an important procedure to extract the water leaving optical signal from the total measurement of the coupled atmosphere and ocean system. Atmospheric correction algorithms in part estimate the aerosol path radiance as well as the ocean surface reflectance and remove them from the total signal. The remaining water leaving signal is due to absorption and scattering inside the water body, which can be used to retrieve the optical properties of seawater constituents and infer their associated biogeochemical conditions (Mobley et al., 2016). Due to the small percentage of the water leaving

signals in the total measurement (Zhai et al., 2017), atmospheric correction requires precise evaluation of the aerosol and ocean surface contributions, which is very challenging when absorbing aerosols are present and when water leaving signals in the near infrared spectral region are non-negligible, both of which are often the case for coastal waters (Sathyendranath, 2000; Wang, 2010).

    Multi-angle, multi-spectral polarimeters (hereafter simply refered to as polarimeters) measure signals that contain rich in-

formation on aerosols and hydrosols (Chowdhary et al., 2005; Hasekamp et al., 2011; Knobelspiesse et al., 2012; Chowdhary et al., 2012; Xu et al., 2016; Stamnes et al., 2018). The aerosol properties obtained from polarimeter data can be explored to improve the atmospheric correction for complex atmosphere and ocean systems. In the Decadal Survey for Earth Science and Applications from Space proposed by National Academy of Sciences for the year of 2017-2027, a polarimetric imager is one of the top priority systems for aerosol observations(National Academies of Sciences, Engineering, and Medicine, 2018).

Meanwhile, National Aeronautics and Space Administration (NASA) plans to launch the Plankton, Aerosol, Cloud and ocean Ecosystem (PACE) mission in the 2022-2023 timeframe (PACE, 2012), which will carry the Ocean Color Instrument (OCI), a hyperspectral radiometer with continuous spectral coverage from the ultraviolet (350 nm) to near-infrared (890 nm), plus a set of discrete shortwave infrared bands (940, 1038, 1250, 1378, 1615, 2130, and 2260 nm). In addition, PACE will carry two polarimeters: the HyperAngular Rainbow Polarimeter (HARP-2) (Martins et al., 2014) and the Spectropolarimeter for Plane-

tary EXploration (SPEXone) (Snik et al., 2010). With this three-instrument payload, PACE will provide new opportunities to perform better atmospheric correction to the OCI imagery with the aerosol information retrieved by the co-located polarimeter measurements.

    To extract the rich information contained in polarimeter measurements, several joint retrieval algorithms have been developed to determine aerosol and water optical properties simultaneously. Oceanic optical properties are usually soley parameterized

by the concentration of the photosynthetic pigment chlorophyll-a ([Chla])(Chowdhary et al., 2005; Hasekamp et al., 2011; Xu et al., 2016; Stamnes et al., 2018). Gao et al. proposed a joint retrieval approach for a coupled atmosphere and ocean system that employs a generalized bio-optical model for coastal waters (Gao et al., 2018). There are seven free parameters in this bio-optical model that describe the absorption and scattering characteristics of different components such as water, phytoplankton, colored dissolved organic matter (CDOM) and non-algal particles (NAP). This retrieval algorithm was validated with synthetic data



generated by a radiative transfer model (Zhai et al., 2009, 2010, 2015, 2017, 2018), which demonstrated high accuracy in the retrieval of water leaving signals and aerosol micro-physical properties for a large variety of atmospheric and ocean conditions. The purpose of this paper is to further validate the algorithm by applying it to airborne observations. Specifically, the retrieval algorithm processes the polarimeter measurements over both open and coastal waters and generates water leaving signals as

well as aerosol properties, which are then compared with in situ measurements to evaluate the accuracy and uncertainties.

In order to accurately fit the field measurements, the original algorithm in Gao et al. (2018) has been further upgraded to include trace gas absorption and an updated instrument noise model. A [Chla]-based bio-optical model has also been added to constrain the water-leaving radiance for open waters, while the general seven-parameter bio-optical model is still used for coastal waters. In this study, both the two bio-optical models are applied over coastal waters in order to evaluate their impacts

on the water leaving signal retrieval. This work builds upon the studies of RSP, AirMSPI and PARASOL (Chowdhary et al., 2005; Hasekamp et al., 2011; Knobelspiesse et al., 2012; Chowdhary et al., 2012; Xu et al., 2016; Stamnes et al., 2018), and extends the retrieval of ocean optical properties from such instruments to coastal regions.

The paper is organized in six sections: Sec. 2 will introduce the data from field measurements used in the retrieval study; Sec. 3 will review the retrieval algorithm; Sec. 4 presents the retrieval results; Sec. 5 discusses the results; and Sec. 6 summarizes

the conclusions.

## 2   Data

In this work, we have applied the joint retrieval algorithm to the measurements acquired by the airborne Research Scanning Polarimeter (RSP) (Cairns et al., 1999; Knobelspiesse et al., 2019). RSP includes six boresighted refractive telescopes that formed three pairs with each pair measuring three spectral bands (Cairns et al., 1999). Nine wavelengths are measured with the

central wavelengths and band width at visible (VIS) bands: 410 (30), 470 (20), 550 (20) and 670 (20)nm, near infrared (NIR) bands: 865 (20) and 960 (20) nm, and shortwave infrared (SWIR) bands: 1590 (60), 1880 (90) and 2250 (120) nm. The scanning directions relative to the instrument baseplate is within $\pm 60°$ with 152 angles and an instantaneous field of view (IFOV) of 14mrad ($0.8°$), which can be geolocated to provide hyperangular measurments of the same target. For the measurements in our following discussion as summarized in Table 1, the spatial resolution is about 100 meters which can be estimated from the

product of the IFOV and aircraft altitude.

Within each pair of the telescopes, one makes measurements of the polarization components at the orthogonal plane of $0°$ and $90°$ denoted as $I_{0°}$ and $I_{90°}$, and the other telescope simultaneously measures the polarization components at $45°$ and $-45°$ denoted $I_{45°}$ and $I_{-45°}$. The polarized measurement is denoted using a Stokes vector $\mathbf{I}_t = (I_t, Q_t, U_t, V_t)^T$, where $Q_t = I_{0°} - I_{90°}$, $U_t = I_{45°} - I_{-45°}$, and $V_t$ is usually negligible for the atmospheric studies. The total radiance used in this

study is an averaged of the radiance remeasured by the two telescopes and is defined as $I_t = (I_{0°} + I_{90°} + I_{45°} + I_{-45°})/2$. The corresponding instrument noise model is provided in (Knobelspiesse et al., 2019) and summarized in Appendix C.

The measurements from two field campaigns are chosen for this study, namely the Ship-Aircraft Bio-Optical Research (SABOR) experiment and the North Atlantic Aerosols and Marine Ecosystems Study (NAAMES). The SABOR experiment was




conducted from July 17th to Aug 7th in 2014 (NASA SABOR webpage), and the NAAMES campaign is a multi-year study where four month-long expeditions took place between 2015 and 2018(NASA NAAMES webpage). During both campaigns the airborne measurements from RSP and in situ measurements from the ocean vessels were acquired. Due to the difficulty of finding polarimeter observations in cloud free conditions over the ocean with coordinated in situ water leaving signal measure-

5  ment, only four cases from SABOR and NAAMES are investigated in this study. Each case is given a name for our discussion by combining its campaign name and the water types: SABOR-Open, SABOR-Coastal, NAAMES-Open, NAAMES-Coastal. The basic information for the measurements including the time, location and instrument geometries are summarized in Table 1. The aerosol optical depth (AOD) from these cases ranges from 0.05 to 0.35. The corresponding RSP files are listed in Appendix A. The locations and polar graphs of the solar direction and the RSP scanning direction for each case are summarized

10  in Figures 1 and 2.

**Table 1.** Summary of datasets from SABOR and NAAMES field campaigns. Case name is given as a combination of the campaign name and water types. The time range is for the start and end time of the corresponding RSP scene. The retrieval time, latitude, longitude, solar and scattering geometry are for the RSP measurements in the corresponding field campaign. The time for the in situ measurement is also given for comparison. All time is in UTC. The altitude is the height of the aircraft which carried RSP. The relative azimuth angle is the relative angle between the RSP scanning direction and the principal plane formed by the solar direction and the zenith direction.

| Case Name | SABOR-Open | SABOR-Coastal | NAAMES-Open | NAAMES-Coastal |
|---|---|---|---|---|
| Date | 07/27/2014 | 07/30/2014 | 05/26/2016 | 11/04/2015 |
| Campaign | SABOR | SABOR | NAAMES | NAAMES |
| Water type | Open | Coastal | Open | Coastal |
| RSP Time Range | [14.183, 14.297] | [15.187, 15.328] | [15.089, 15.383] | [18.347, 18.432] |
| RSP Retrieval Time | 14.231 | 15.249 | 15.129 | 18.416 |
| In Situ Measurement Time | 19.77 | 17.95 | 14.33 | N/A |
| Time Zone | UTC-4 | UTC-5 | UTC-3 | UTC-5 |
| Latitude | 36.651° | 36.915° | 47.089 | 39.181 |
| Longitude | -67.426° | -75.796° | -37.751 | -75.241 |
| Altitude | 8.99km | 8.87km | 6.70km | 6.76km |
| Solar zenith | 35.7° | 31.2° | 27.0° | 59.4° |
| Relative azimuth | 60° | 32° | 83° | 75° |
| Scattering angle range | [103.3°,148.3°] | [88.5°,164.1°] | [116.1°,154.1°] | [90.7°,122.8°] |



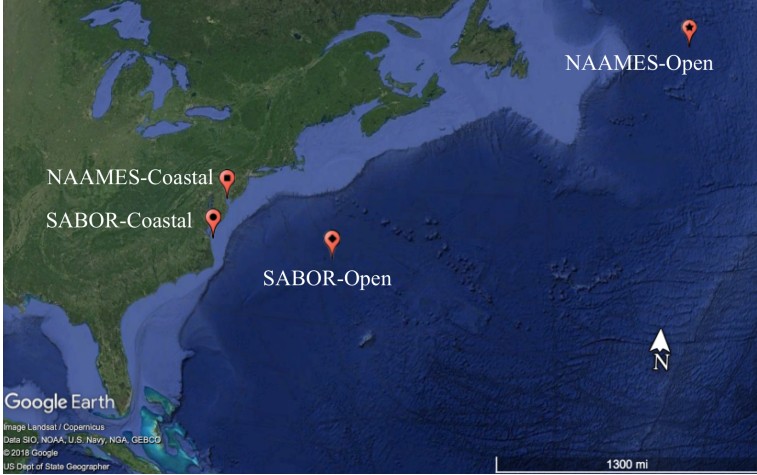

**Figure 1.** The locations of the RSP measurements as listed in Table 1.

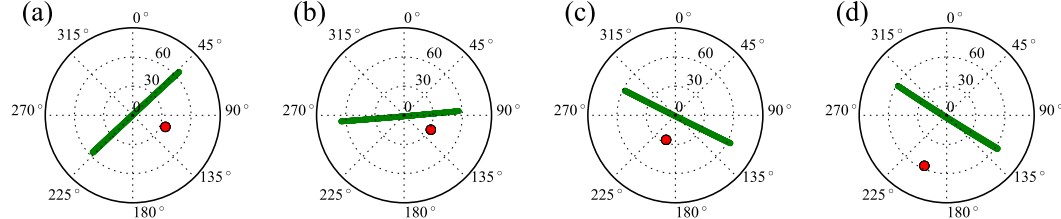

**Figure 2.** The polar plot for solar direction (red dot) and the RSP scanning direction (green line) for each case listed in Table 1: (**a**) SABOR-Open, (**b**) SABOR-Coastal, (**c**) NAAMES-Open, and (**d**) NAAMES-Coastal. The north direction is chosen as azimuth angle zero.

## 3 Algorithm and methodology

The joint retrieval algorithm for atmospheric correction is based on the multi-angle, multi-wavelength, and polarization measurements acquired by RSP. In the following, we will first introduce the definition of the measurement and retrieval quantities. The retrieval algorithm is an optimization approach that minimizes the difference between the RSP measurements and the forward model simulations, formally defined as the cost function in Eq. 3 below. The forward model is the radiative transfer model that computes the reflectance at sensor level using both aerosol and ocean bio-optical models as reviewed in the previous study (Gao et al., 2018; Zhai et al., 2010).



### 3.1 Reflectance and remote sensing reflectance

Using the measured Stokes vector components, the total reflectance $\rho_t$ and polarized reflectance $\rho_P$ at sensor level are defined as

$$\rho_t = \frac{\pi r^2 L_t}{\mu_0 F_0}, \quad \rho_P = \frac{\pi r^2 \sqrt{Q_t^2 + U_t^2}}{\mu_0 F_0} \tag{1}$$

where $F_0$ is the extraterrestrial solar irradiance, $\mu_0$ is the cosine of solar zenith angle, and $r$ is the solar distance in astronomical units. The total reflectance $\rho_t$ includes contributions from the molecular (Rayleigh) scattering $\rho_R$, aerosol scattering $\rho_a$, the interaction term of Rayleigh and aerosol $\rho_{Ra}$, surface reflectance such as sunglint $\rho_g$ and whitecaps $\rho_{wc}$, as well as the water leaving contributions $\rho_w$. In ocean optics literature, L is often used to denote radiance which is the same as I in a Stokes vector(Mobley, 1994). The objective of the atmospheric correction is to obtain $\rho_w$ by removing all other contributions–this

requires accurate modeling of the molecular and aerosol scattering and the surface reflectance.

Remote sensing reflectance, defined as $R_{rs} = L_w^+/F_d^+$, is commonly used to represent the water leaving signal originating from scattering from the water body, where $L_w^+$ is the upwelling water-leaving radiance just above the water surface after the atmospheric correction and $F_d^+$ is the downwelling irradiance just above the water surface. The superscript +/- is used to denote just above/below the ocean surface. The nadir direction is used to compute the remote sensing reflectance. The observed

water leaving reflectance at the airborne or spaceborne sensor is denoted as $\rho_w^{Sensor} = \pi t_u L_w^+/[\mu_0 F_0]$, which represents the water leaving reflectance just above ocean surface transmitted to the sensor through a diffuse transmittance $t_u$. $\rho_w^{Sensor}$ can be obtained from the total reflectance measured at the sensor by removing the contribution from molecular and aerosol path radiance, ocean surface reflectance (e.g., sunglint, white caps) and their interaction terms (Gao et al., 2018). The remote sensing reflectance can be related to the water leaving reflectance as

$$R_{rs} = \frac{\rho_w^{Sensor}}{\pi t_d t_u} \tag{2}$$

where $t_d$ is the same as $t_u$ but represents the downward transmittance of the solar irradiance to the water surface (Gao et al., 2000). This definition is used in our study to conduct the atmospheric correction and calculate the remote sensing reflectance. A detailed mathematical treatment is in Appendix B.

### 3.2 Retrieval algorithm

An optimization approach is used to retrieve the aerosol and ocean optical properties, where the measured reflectance are compared with the reflectance computed from a forward model using a set of parameters that specify the aerosol and ocean optical properties. If the agreement is within a pre-defined criterion, the optimization procedure finishes, otherwise, the retrieval parameters are perturbed and the whole process iterates until the convergence criterion is satisfied. The optimization algorithm used in this study is the Levenberg-Marquet method(Moré et al., 1980). A least square cost function is defined to quantify the

difference between the measurement and the simulation from a forward model as:

$$\chi^2(\mathbf{x}) = \frac{1}{N} \sum_i \left( \frac{[\rho_t(i) - \rho_t^f(\mathbf{x};i)]^2}{\sigma_t^2(i)} + \frac{[\rho_P(i) - \rho_P^f(\mathbf{x};i)]^2}{\sigma_P^2(i)} \right), \tag{3}$$



where $\rho_t$ and $\rho_P$ are the measured reflectance defined in Eq.1, $\rho_t^f$ and $\rho_P^f$ denotes the reflection simulated from a forward model specified by a parameter vector $\mathbf{x}$, $i$ indicates the measurement at different angles and wavelengths, and N is the total number of the measurements used in the retrieval. The total uncertainties of the reflectance and the polarized reflectance are denoted as $\sigma_t$ and $\sigma_P$ (Knobelspiesse et al., 2019). The total uncertainty includes the instrument measurement uncertainties as

discussed in Appendix C, the variance from averaging nearby RSP pixels (5 pixels are used in this study, which corresponds to a surface pixel size of approximately 500 meters), and the modeling uncertainties with an estimated percentage error similar to the measurement uncertainty. More details in the retrieval algorithm were discussed in Gao et al.(Gao et al., 2018).

The forward model in the retrieval algorithm describes radiative transfer in the coupled atmosphere and ocean system. The atmosphere and ocean system are divided into three layers, with a top molecular layer, a middle layer filled by a mixture of

molecules and aerosols and then an ocean layer with a rough water interface. The aerosol top height is assumed to be 2km in this work. The aerosol and ocean surface representations are summarized in Table 2, where the aerosol volume distribution is represented as the summation of six size modes with three sub-modes of fine mode aerosols and another three sub-modes of coarse mode aerosols (Gao et al., 2018). The complex aerosol refractive index spectra for fine and coarse mode are represented by the principle component analysis (PCA) of aerosol refractive index spectral measurements(Shettle and Fenn, 1979; d'Almeida

et al., 1991). Only the major spectral variation represented by the first order of the principle components is considered(Gao et al., 2018). In the study of the SABOR-Coastal case, where the aerosol loading is relatively large, we also compare the results between the PCA representation and a more flexible representation of combining PCA with small adjustments in the refractive indices for the wavelength of 410nm and 470nm in order to assess the possible cause of the bias at shorter wavelengthes. Implementation details of the refractive index adjustment will be discussed with the SABOR-Coastal case. Moreover, in order

to model the field measurement, the previous forward model (Gao et al., 2018) is further developed in this study by including the gas absorption due to ozone, oxygen, water vapor, nitrogen dioxide, methane, and carbon dioxide (Zhai et al., 2018). The aerosol scattering and absorption properties are then mixed with the gas absorption within the molecular and aerosol mixing layer.

**Table 2.** The forward model for aerosol refractive index, volume distribution and ocean surface.

| Component | Model | Parameters |
|---|---|---|
| Aerosol volume distribution | Six sub-modes | Volume density of each mode |
| Aerosol refractive index spectra | Principle component analysis(PCA) | PCA coefficients |
| Ocean surface | Cox-Munk model(Cox and Munk, 1954) | Wind speed (scalar) |

Bio-optical models can be used to describe the scattering and absorption of the key constituents in ocean waters including

pure water, phytoplankton, CDOM and NAP (Mobley, 1994). The pure sea water absorption and scattering coefficients($a_w$,$b_w$) are obtained from measurements(Kou et al., 1993; Pope and Fry, 1997; Zhang and Hu, 2009), and the pure sea water phase function $P_w$ is similar to Rayleigh scattering (Mobley, 1994). To model the coastal water optical properties, our bio-optical model considers seven parameters that explicitly define the scattering and absorption properties from phytoplankton, CDOM





ad NAP(Gao et al., 2018). The key absorption and scattering properties are summarized in Table 3, which includes the absorption coefficients of phytoplankton ($a_{ph}$), the total absorption coefficient of CDOM and NAP($a_{dg}$), the total particulate backscattering coefficient($b_{bp}$) for both phytoplankton and NAP, and the total particulate backscattering fraction $B_p$. $a_{ph}$ is a function of $[Chla]$ with coefficients $A_{ph}$ and $E_{ph}$ provided in Bricaud et al. (1998). The particulate phase function is described

by the the Fouriner-Forland phase function (FF), which is an analytical function that can be determined by the backscattering fraction of $B_p$ (Fournier and Forand, 1994). To obtain the total Mueller matrix of water, the FF phase function is mixed with the water phase function $P = (P_w b_w + b_{bp} FF)/(b_w + b_{bp})$, then multiplied by the normalized Mueller matrix derived from measurements (Voss and Fry, 1984; Kokhanovsky, 2003) where the polarization properties are assumed to be invariant.

**Table 3.** The generalized ocean bio-optical model (Bio-2) for coastal waters.

| Component | Model | Parameters |
|:---:|:---:|:---:|
| $a_w, b_w$ | Measurements (Kou et al., 1993; Pope and Fry, 1997; Zhang and Hu, 2009) | N/A |
| $P_w$ | Rayleigh like scattering (Mobley, 1994) | N/A |
| $a_{ph}$ | $A_{ph}(\lambda)[Chla]^{E_{ph}(\lambda)}$ | $[Chla]$ |
| $a_{dg}$ | $a_{dg}(440)\exp[-S_{dg}(\lambda - 440)]$ | $a_{dg}(440), S_{dg}$ |
| $b_{bp}$ | $b_{bp}(660)(\lambda/660)^{-S_{bp}}$ | $b_{bp}(660), S_{bp}$ |
| $B_p$ | $B_p(660)(\lambda/660)^{-S_{Bp}}$ | $B_p(660), S_{Bp}$ |
| $P_p$ | Fournier-Forland phase function(FF) (Fournier and Forand, 1994) | $B_p$ |

When studying open waters, it is often assumed that $[Chla]$ can be used as a single parameter to describe the optical

properties of all seawater constituents (Chowdhary et al., 2012; Xu et al., 2016). For open waters, we therefore constrain the parameters in the previously described bio-optical model using only [Chla]. Specifically, the parameters describing $a_{dg}(440)$, $S_{dg}$, $b_{bp}(660)$, $S_{bp}$ and $B_p$ are re-specified in terms of [Chla] as shown in Appendix D. It is assumed that no contribution from NAP is significant in open ocean waters. In practice, we use the [Chla]-based bio-optical model (Bio-1) in the open ocean to reduce uncertainties associated unnecessarily with multiple parameters, while we use the full seven parameter bio-

optical model (Bio-2) in coastal waters. We then evaluate the difference in using both the two bio-optical models for coastal water studies in order to understand the applicability of the different model parameterizations. We acknowledge that alternate parameterizations exist, but a detailed exploration of them exceeds the scope of this paper. Furthermore, if there is *a priori* knowledge of the parameters in the generalized bio-optical model, the number of retrieval parameters can be reduced by assuming pre-specified values. For example, a similar bio-optical model for $a_{dg}$ and $b_{bp}$ has been proposed in a spectral

optimization approach (Kuchinke et al., 2009), where the spectral coefficient $S_{dg}$ and $S_{bp}$ are assumed to be known from existing studies. The reduced number of free parameters may help reduce uncertainties in the retrieved quantities.




## 4    Joint Retrieval Results

The retrieval algorithm discussed in the last section is applied to the RSP data acquired in the SABOR and NAAMES campaigns. Two locations are selected in each campaign: one for open ocean waters and the other for coastal ocean waters as summarized in Table 1. The [Chla]-based bio-optical model (Bio-1) is applied to the open water cases, while both the [Chla]-based bio-optical model (Bio-1) and the seven parameter bio-optical model (Bio-2) are applied to the coastal water cases to explore the impact of model parameterization in the atmospheric correction.

For the SABOR measurements, we compared the retrieved aerosol optical depth with the the aerosol product from the High Spectral Resolution Lidar (HSRL) (Hair et al., 2008) and the Aerosol Robotic Network (AERONET) (Dubovik and King, 2000). The collocated in situ measurements of the water leaving signals are compared with the retrieval results from SABOR-Open, SABOR-Coastal, and NAAMES-Open. For NAAMES-Coastal, there are no in situ measurements available; instead we compared with the ocean color product derived from the Moderate Resolution Imaging Spectroradiometer (MODIS) onboard Aqua.

The $\chi^2$ value of a converged case indicates the retrieval quality. A $\chi^2$ close to 1 means that the average difference between the measurement and the simulation are comparable to the uncertainty model quantified by $\sigma_t$ and $\sigma_P$ (Rogers, 2000). If $\chi^2$ is much larger than 1, it may suggest underfitting, where the forward model does not sufficiently describe the measurements. For example, this could indicate that the measurements are influenced by clouds and should be screened. In practice, since the retrievals cannot always reach the global minimum due to the local minima of the cost function, the converged $\chi^2$ value depends on the initial values of the retrieval parameters. In order to explore the corresponding retrieval uncertainties, we ran the retrieval algorithm 50 times for each case listed in Table 1. Each time the initial values of the retrieval parameters are different and randomly generated. The cumulative probability (CP) of all 50 converged $\chi^2$ values is evaluated. The $1\sigma$ uncertainties of the retrieval parameters can be determined by the range of variability of all retrievals with $\chi^2$ smaller than that of CP=70%. Within this CP, the minimum and maximum cost function values are denoted as $\chi^2_{min}$ and $\chi^2_{max}$, corresponding to the best and worst fitted simulations, respectively. For the four cases in our study, the $\chi^2_{min}$ and $\chi^2_{max}$ are summarized in Table 4. The implications of $\chi^2$ values and retrieval uncertainties will be discussed in details for each case in the following sections.





**Table 4.** The minimum and maximum values, $\chi^2_{min}$ and $\chi^2_{max}$, for CP=70% with the two bio-optical models and the four cases listed in Table 1. Bio-1 is applied for open waters, while both Bio-2 and Bio-1 are applied for coastal waters. All cases use the seven RSP bands except for the one indicated by asterisk which did not use SWIR bands.

| Case | Bio-1/Bio-2 | $\chi^2_{\mathbf{min}}$ | $\chi^2_{\mathbf{max}}$ |
|---|---|---|---|
| SABOR-Open | Bio-1 | 1.1 | 5.0 |
| SABOR-Coastal | Bio-2 | 0.9 | 2.7 |
| | Bio-1 | 0.9 | 1.3 |
| NAAMES-Open | Bio-1 | 1.8 | 2.1 |
| | Bio-1* | 0.7 | 1.1 |
| NAAMES-Coastal | Bio-2* | 0.16 | 1.8 |
| | Bio-1* | 19.6 | 25.2 |

## 4.1 SABOR-Open waters: (07/27/2014)

During the SABOR 2014 field campaign, RSP measurements were made from the NASA LaRC's King Air UC-12B aircraft at heights around 9km over the Atlantic region across both open and coastal waters(Ottaviani et al., 2018). HSRL was also on-board the aircraft, providing accurate aerosol optical depth information that is useful to validate the retrieved aerosol properties from our model. Coordinated in situ measurements from the R/V Endeavor provided water leaving reflectance at various locations for both open and coastal waters. On July 27, 2014, the vessel for the SABOR-Open case was located near 700 km away from the coast as shown in Figure 1. The in situ measurement of water leaving signals were collected using a Satlantic HyperPro tethered in buoy mode(Chase et al., 2017). In this study we compared our retrieval results with these HyperPro measurements, all of which are available from NASA's SeaBASS (NASA SeaBASS webpage). The upwelling radiance $L_u$ is measured at a depth of 0.2 meters below ocean surface, and then extrapolated to just below the ocean surface ($L_u^-$). The upwelling radiance just above the water surface $L_w^+$ can be estimated as

$$L_w^+ = \frac{T L_u^-}{n_w^2}, \tag{4}$$

where T is the transmittance from just below the water surface to just above the water surface with a value of 0.98 and the $n_w$ is the water refractive index with a value of 1.34. The remote sensing reflectance is then computed using $L_w^+$ for the comparison with the retrieval results.





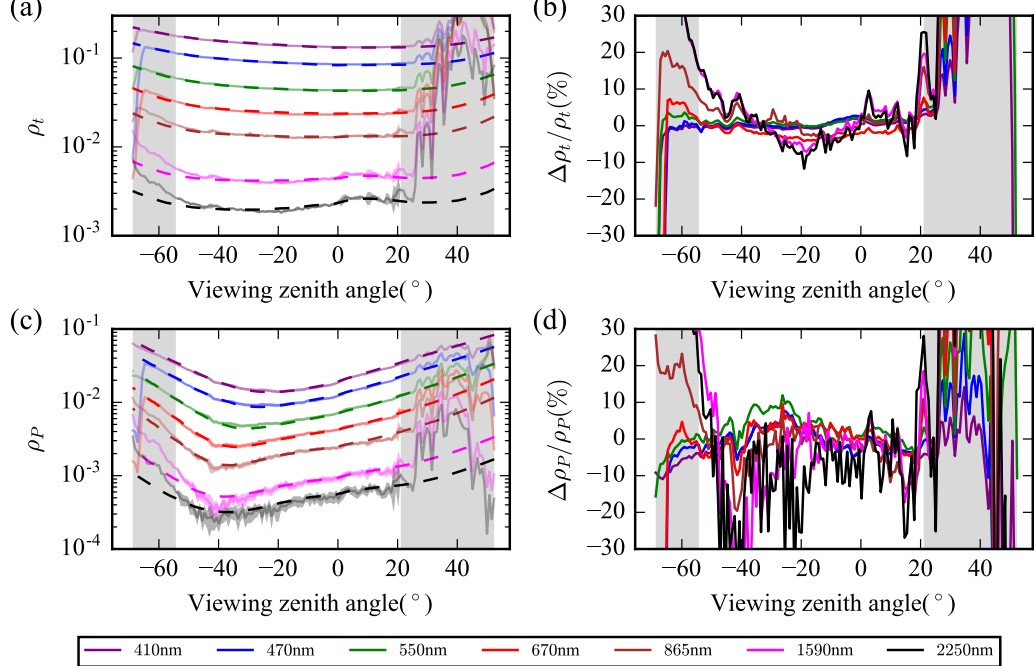

**Figure 3.** (**a**) The comparison of the RSP measurement and simulation reflectance $\rho_t$ for the SABOR-Open case at UTC=14.231, (**b**) the relative differences, (**c**) and (**d**) are the same for (**a**) and (**b**) but for polarized reflectance $\rho_P$. The solid line is the measurement data with vertical line width as measurement uncertainties. Dashed line is the simulation results from the retrieval. The gray area covered angles were not used in the retrieval. The minimum cost function value is $\chi^2_{min} = 1.1$, and the bio-optical model used in this retrieval is the Bio-1 model.

The solar and viewing geometry is summarized in Table 1 for SABOR-Open case and is also shown in the polar plot of the geometry in Figure 2 with a solar zenith angle of 35.7°. The RSP viewing directions are away from the principal plane by a relative azimuth angle of 60° on average. As shown in Fig. 3, the measured reflectance does not contain prominent sunglint reflection peak. The solid lines with a vertical spreading indicate the measurement with uncertainties. A portion of directions

5    are influenced by clouds that are masked out in gray color in Fig. 3 and excluded from the retrieval.

The retrieval algorithm with Bio-1 is applied on the measurements as indicated by the solid line in 3 (a) and (c). The maximum cost function value is $\chi^2_{max} = 5.0$. The corresponding retrieval uncertainties for AOD and remote sensing reflectance are calculated as discussed previously. The best fitted simulation result is shown in Figure 3 (a) and (c) by the dashed line with $\chi^2_{min} = 1.1$. The percentage difference between the measurement mean value and the simulation results are shown for both

10    reflectance and polarized reflectance ($\rho_t$ and $\rho_P$) in 3 (b) and (d). Among most angles, the percentage difference for $\rho_t$ is less than 5%, but there are slightly larger percentage errors up to 10% for SWIR bands at a few angles. For $\rho_P$, the overall percentage difference is less than 10%, except for the SWIR bands where the largest percentage difference around $-40°$ can go beyond 30% due to the small polarized reflectance less than $10^{-3}$.





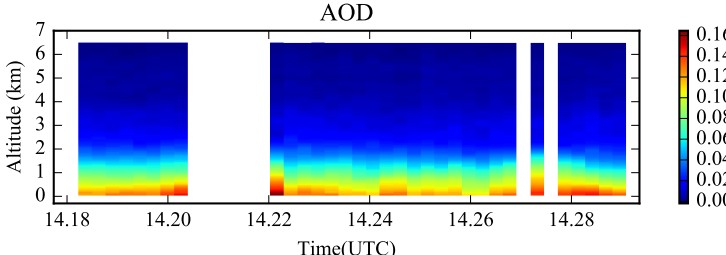

**Figure 4.** The cumulative aerosol optical depth (AOD) from HRSL. The white stripe indicates no HSRL retrieval due the presence of cloud.

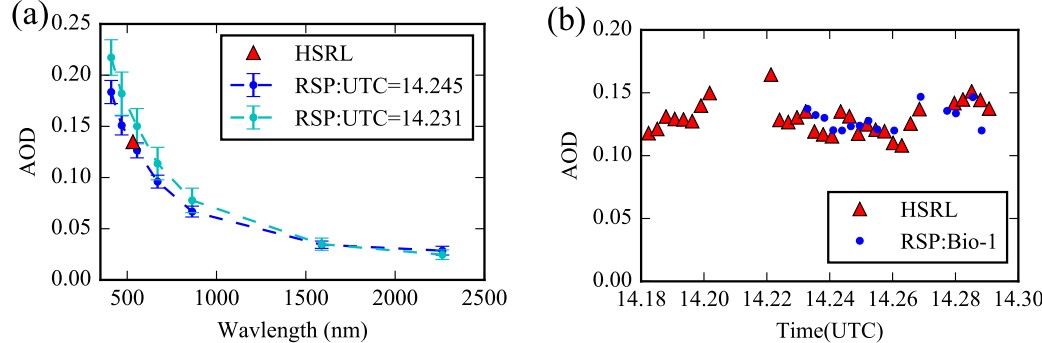

**Figure 5. (a)** The retrieved aerosol optical depth (AOD) from RSP measurement at UTC=14.231 (SABOR-Open) and 14.245 comparing with HSRL AOD at 532nm similar at both time. The error bars indicate the retrieval uncertainties. The retrieval algorithm is based on the [Chla]-based bio-optical model (Bio-1). **(b)** The comparison of the and RSP AOD retrieval and HSRL AOD product.

HSRL provided complementary measurements of the aerosol optical depth, which can be used to validate our RSP retrievals. The vertical cumulative profile of HSRL AOD is shown in Figure 4, where aerosols are mostly located with a vertical region within 1km from the surface. The retrieved aerosol optical depth spectrum from RSP is compared with HSRL optical depth in Figure 5 (a). At UTC=14.231, the HSRL AOD is 0.135 while the averaged RSP AOD is 0.150 with a $1\sigma$ uncertainty of 0.024.

5    The RSP AOD is larger than the HSRL AOD from HSRL by 0.015. This is possibly due to the different viewing geometry from these two instruments. HSRL observes a vertical profile of the aerosols as shown in Figure 4, while RSP observes multiple viewing angles around $\pm 60°$ relative to the instrument base plate. At UTC=14.221, a location 4.86km away the SABOR-Open case, HSRL AOD is larger with a value of 0.16 as shown in Figure 5 (b), which may contribute to the different RSP observed AOD. Moreover, a nearby cloud may still influence in the remaining angles of the RSP measurement through multiple scattering even after masking the obvious cloud impacted region. To assess this hypothesis, we considered a location at UTC=14.245,

10    which is further away from the SABOR-Open case by 6.66km. Here, the HSRL AOD is the same as the SABOR-Open case with a relatively clean and smooth variation in the nearby region as shown in Figure 4. The retrieved aerosol optical depth at





532 nm has a better agreement with the HSRL AOD as shown in Figure 5 (a). The retrieval uncertainties reduce from 0.021 to 0.011.

Using the averaged retrieved aerosol properties at UTC=14.245 as the initial value, the retrieval algorithm is applied to the RSP measurement along the flight track. Figure 5 (b) shows the comparison between the RSP and HSRL AOD, which demonstrates consistency. No RSP retrieval is shown for UTC 14.18 to 14.22 due to the large influence of cloud in the measurement.

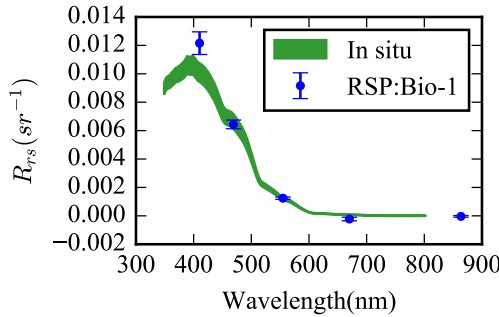

**Figure 6.** The comparison of the RSP retrieved remote sensing reflectance with the in situ measurements for SABOR-Open 07/30/2014. The vertical line width indicates the uncertainties of the in situ measurements, while the error bars indicate the retrieval uncertainties.

The retrieved remote sensing reflectance is compared with the in situ measurement as shown in Figure 6. The $1\sigma$ uncertainty of the in situ measurement is indicated by the vertical line width, which was calculated using the signal variability within the 5 minute measurement duration. The RSP measurement was made at UTC=14.231 and the in situ measurement was made at UTC=19.77 as summarized in Table 1. The distance between these two locations is less than 0.1km. The vertical bar indicates the RSP retrieval uncertainties. The maximum remote sensing reflectance obtained from the in situ measurement is 0.0106, while the $R_{rs}$ from the RSP retrieval has a peak at 410nm with a value of 0.0122. The retrieval uncertainties at 410nm has a value of 0.00080, while the uncertainty for 470nm is 0.00031, and other bands less than $10^{-4}$. Figure 6 shows that our remote sensing reflectance agrees with the in situ measurements for all wavelength bands longer than 470 nm. At 410 nm, the difference is the largest, which is however acceptable due to inherent retrieval uncertainty associated with large reflectance signal and the possible small scale variability of ocean optical properties at deep blue wavelengths.

## 4.2  SABOR-Coastal waters (07/30/2014)

On July 30, 2014 during the SABOR campaign, R/V Endeavor was located 20km away from the coast with in situ measurements available as shown by the SABOR-Coastal location in Figure 1. We executed the joint retrieval of the aerosol properties and water leaving reflectance using both the [Chla]-based bio-optical model (Bio-1) and the seven parameter bio-optical model (Bio-2). The retrieved properties are compared with the in situ measurement from HyperPro and the AOD product from HSRL and AERONET.





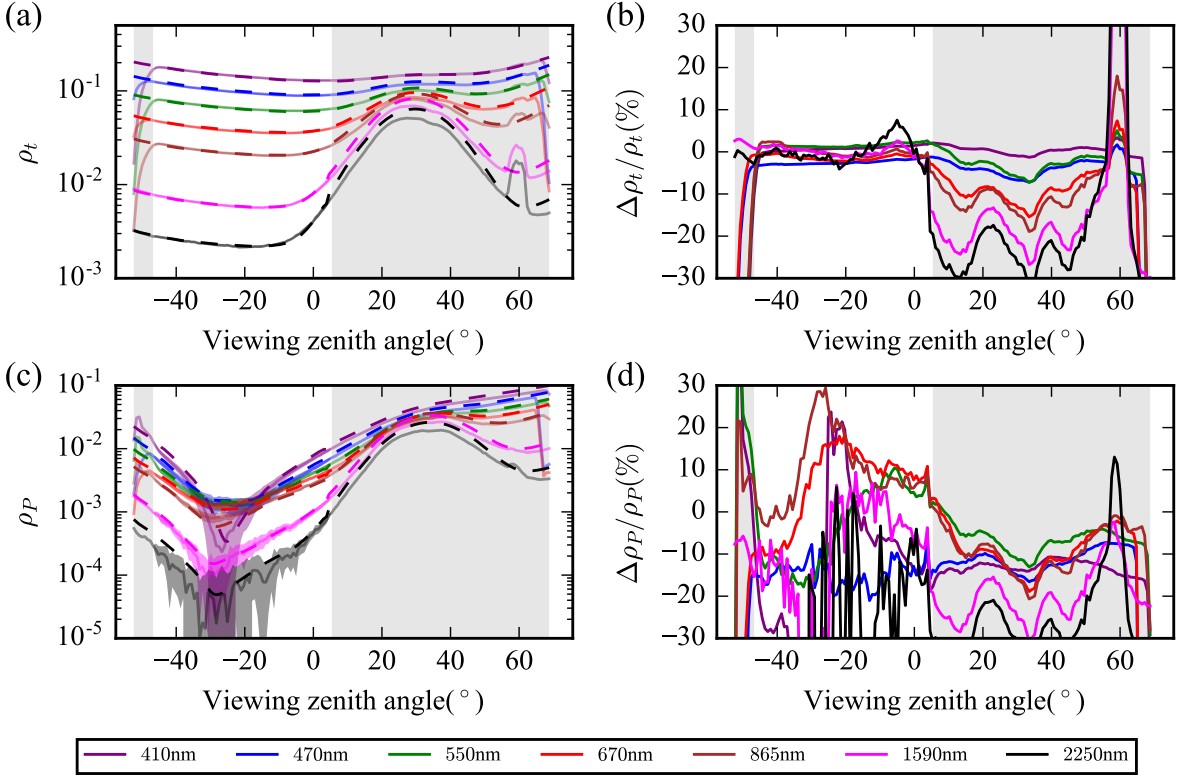

**Figure 7.** Same as Figure 3 but for SABOR-Coastal 07/30/2014. The minimum cost function value is $\chi^2_{min} = 0.9$ and the bio-optical model used here is Bio-2.

The solar zenith angle is 31.2° and the relative viewing azimuth is 32° between the RSP scanning direction and principal plane for the SABOR-Coastal case, as shown in Figure 2 (b) and in Table 1. Figure 7 shows the comparison between the RSP measured and model fitted polarized reflectance field. As shown in Figure 7 (a), the sunglint is prominent in the measurement data. A test retrieval with the sunglint considered produces a larger aerosol optical depth and larger aerosol absorption as compared with AERONET AOD. This suggests that the retrieval optimization decreases the direct light while retaining similar scattering signals. Moreover, if the sunglint is removed as shown in the gray area in Fig. 7, the retrieval bias is greatly reduced. Figure 7 (b) shows the retrieval results without considering the contribution of the sunglint matches well in $\rho_t$ for wavelength 410nm, 470nm and 550nm for viewing zenith angle between 0 and -50°.

The maximum cost function values are $\chi^2_{max} = 2.7$ and 1.3 for Bio-2 and Bio-1, respectively, indicating smaller uncertainties when using Bio-1. Figure 7 shows the comparison of the measurement and best fitted simulation results using Bio-2 with $\chi^2_{min} = 0.9$. There are less than 3% of percentage differences between the measured and simulated $\rho_t$ for all the wavelength and most angles. Meanwhile there are relatively large percentage difference for $\rho_P$ between the measurement and simulation,





especially in the backscattering direction for the SWIR bands as shown in Figure 7 (c) and (d), but the uncertainties in the measurement are also larger, which reduce their influence in the cost function.

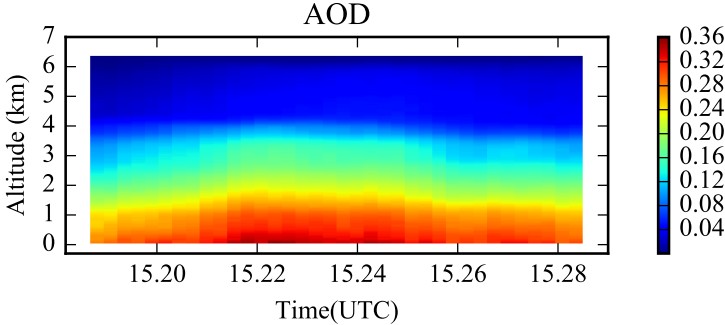

**Figure 8.** The cumulative aerosol optical depth (AOD) from HSRL.

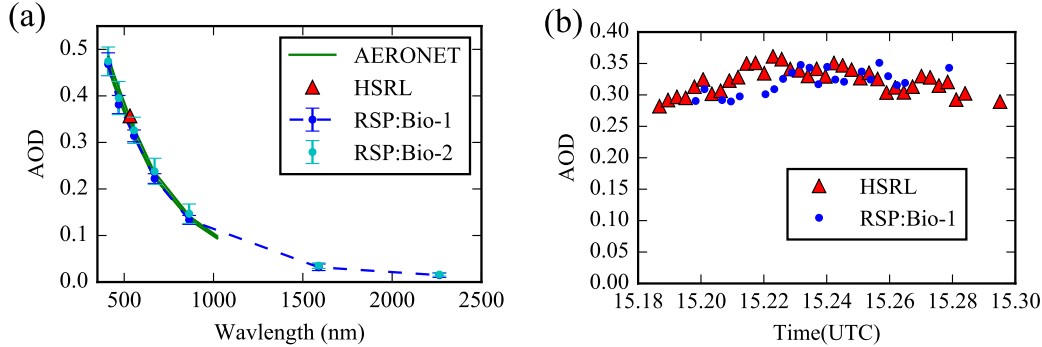

**Figure 9.** (**a**) The comparison of the RSP retrieved AOD, the HRSL AOD at 532nm and the AOD from AERONET site (COVE_SEAPRISM) for SABOR-Coastal. The error bars indicate the retrieval uncertainties. (**b**) The comparison of the RSP retrieved AOD at 550nm with the HSRL AOD at 532nm across the flight track.

The vertical profile of HSRL AOD along the flight track is shown in Figure 8, which indicates a small variation of the AOD and no apparent influence from clouds. Figure 9 shows both bio-optical models can achieve accurate AOD retrieval as compared with the HSRL AOD, and the AOD spectrum from the nearby AERONET site (COVE_SEAPRISM) with a distance of about 9.4km. The COVE_SEAPRISM site measures AOD through the direct sun light extinction from a CIMEL-based system, called the Sea-Viewing Wide Field-of-View Sensor (SeaWiFS) Photometer Revision for Incident Surface Measurements (SeaPRISM), at eight wavelength of 412nm, 443nm, 490nm, 532nm, 551nm, 667nm, 870nm, and 1020nm (Zibordi et al., 2009). At wavelength of 550nm, RSP retrievals using Bio-1 obtain AOD=$0.314 \pm 0.013$, while the retrievals using Bio-2 produce AOD=$0.326 \pm 0.028$. The RSP AOD at 550nm retrieved from both Bio-1 and Bio-2 are comparable with the HSRL AOD of 0.340 at 532nm. Although the RSP measurements are over coastal waters, the results using Bio-1 have a smaller





uncertainties compared with Bio-2, probably resulting from the use of fewer retrieval parameters. The seven-parameter bio-optical model may be unnecessary in this case due to the small water leaving signal and the large aerosol contribution. In the NAAMES-Coastal case that we will discuss later, Bio-2 has to be employed to achieve convergence because the water leaving signal is strong and the aerosol contribution is weak. Figure 9 (b) shows the RSP AOD retrieval with Bio-1 agrees well with

the HSRL AOD along the track with $\chi^2_{max} = 5.0$. When using Bio-2, there are fewer retrieval results to reach the similar cost function level (data not shown) for the same set of initial values.

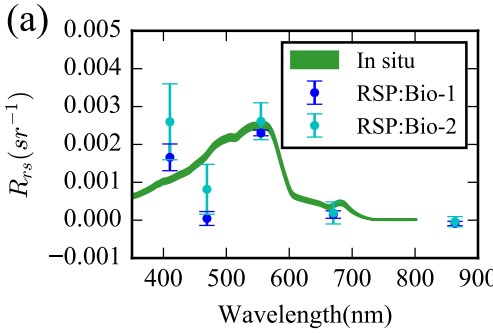 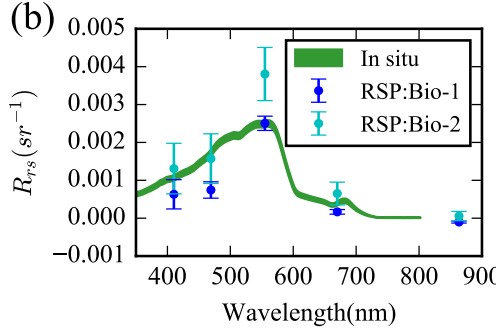

**Figure 10.** (**a**) The comparison of the remote sensing reflectance from RSP retrieval and the in situ measurement using two bio-optical models: Bio-1 and Bio-2. (**b**) same as (**a**) but with extra retrieval parameters which adjust both the real and imaginary parts of the fine mode refractive index at 410nm and 470nm. The vertical line width indicates the uncertainties of the in situ measurements, while the error bars indicate the retrieval uncertainties.

The retrieved remote sensing reflectance is compared with the in situ measurements from HyperPro, which is 1.7km away from RSP measurements for the SABOR-Coastal case. The retrieved $R_{rs}$ shares similar spectral shape for the two bio-optical models, but with different uncertainties as shown in Figure 10 (a). For example, Bio-1 retrieves $R_{rs}$ at 410nm with a value

of $0.0017 \pm 0.00035$, while Bio-2 obtains $R_{rs}$ at 410nm with a value of $0.0026 \pm 0.001$, but both overestimate the in situ measurement value of 0.0010. At 470nm, both retrievals underestimate the in situ measurement, with Bio-2 slightly closer to the in situ observations. The wavelength at 550nm $R_{rs}$ is more accurately retrieved with uncertainties smaller than $1.0^{-4}$. The difference in the $R_{rs}$ retrieval compared with the in situ measurement may be due to the small magnitude of the water leaving signals and the large aerosol loadings.

Furthermore, there may be small variations in the aerosol refractive index spectrum that are not captured by the smooth representation of the PCA, which may affect the retrieval of water leaving radiance adversely. To explore the possibility of achieving better water leaving radiance retrieval by accounting for this variation, we conducted the retrieval again by adding four retrieval parameters as the perturbations to the real and imaginary parts of the PCA refractive indices at 410nm and 470nm. The perturbations of the real parts are within $\pm 0.1$ and of the imaginary parts are within $\pm 0.01$. A better agreement

of the spectral shape of the retrieved $R_{rs}$ can be found for both bio-optical models as shown in Figure 10 (b). It should be noted that the SABOR-Coastal is the only case which needs the adjustment of refractive index at deep blue wavelengths. A





larger set of validation dataset is needed to determine the scope of scenes which needs this refractive index adjustment, which is currently unavailable in the community.

### 4.3 NAAMES-Open waters (05/26/2016)

On May 26 during the NAAMES02 field campaign in 2016, the aircraft flew over an open water region that was free from clouds. In situ measurements of water leaving radiance are available from the R/V Atlantis, though they are not well co-located with the RSP measurement (the distance between the RSP footprint and the nearest in situ measurement is about 100 km). Despite the rather larger distance, it is still useful to compare the RSP retrieval and in situ measurement, assuming that the spatial variation in water properties is minimal at such open water, offshore site. The in situ water leaving signal was acquired by the Compact-Optical Profiling System (C-OPS) instrument, which measured the upwelling radiance ($L_u$) and the downwelling irradiance ($E_d$) as a function of depth (NASA SeaBASS webpage). The data was then extrapolated to just above water surface using Eq. 4 to compute the remote sensing reflectance. The in situ measurements were collected at 18 wavelengths: 320, 340, 380, 395, 412, 443, 465, 490, 510, 532, 555, 560, 625, 665, 670, 683, 710, and 780nm, and the data is publicly available in NASA's SeaBASS.

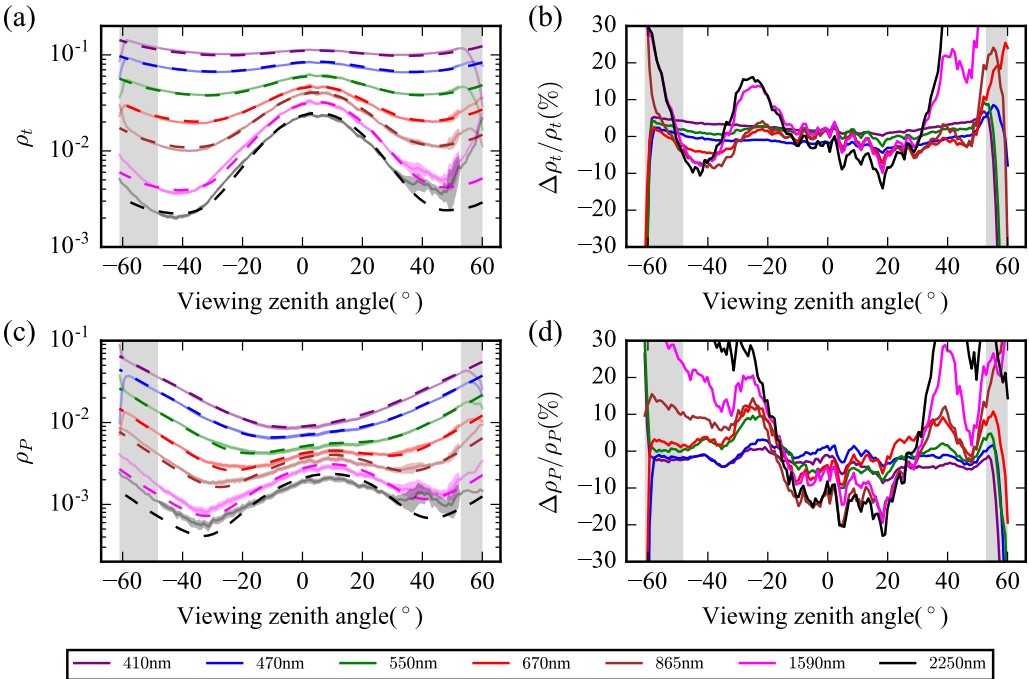

**Figure 11.** Same as Figure 3 but for NAAMES-Open on 05/26/2016. The minimum cost function value is $\chi^2_{min} = 1.8$ and the bio-optical model used here is Bio-1.





As shown in Figure 2 (c) and Table 1, the RSP scanning direction for NAAMES-Open is almost perpendicular to the principle plane. However, the solar zenith angle is 27°, which is quite small. As a result the RSP measurements contain prominent sunglint as shown in Figure 11. The figure shows the best fitted simulation result with $\chi^2_{min} = 1.8$. Both the diffuse reflectance and sunglint have good agreement with the simulated reflectance $\rho_t$, with the percentage error generally less than

5% for VIS bands, though larger errors approch 10% in the NIR bands around the sunglint. There are error even larger than 30% for SWIR reflectance at viewing zenith angle greater than 30°. For the polarized reflectance, the percentage difference is generally less than 10% for VIS bands, but there are more prominent differences at both sides of sunglint especially for the SWIR bands.

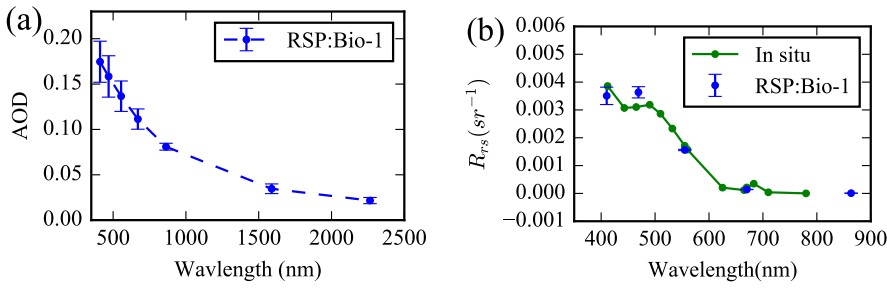

**Figure 12.** (**a**) The RSP retrieved AOD with uncertainties, (**b**) The comparison of the RSP retrieved remote sensing reflectance with in situ measurement. The error bars indicate the retrieval uncertainties.

In order to discuss the retrieval uncertainties, a maximum cost function of $\chi^2_{max} = 2.1$ is obtained. The retrieved optical

depth at 550nm is $0.137 \pm 0.017$ as shown in Figure 12(a). The maximum uncertainties for AOD are at 410nm with a value of 0.009, probably relating to the large AOD at the short wavelength. The remote sensing reflectance can be accurately determined with good agreement comparing with the in situ measurement as shown in Figure 12 (b). The retrieval uncertainties for remote sensing reflectance at band 410nm is 0.00031, which is larger than other bands.

The measurements at the SWIR bands at viewing angles between 30° and 50° show a peak with large uncertainties. These

SWIR data lead to larger aerosol retrieval uncertainties, i.e., excluding the SWIR bands decreases the AOD uncertainties at 550nm from 0.017 to 0.0084. The cost function decreases from 1.8 to 0.7 if the SWIR bands are excluded. However excluding the SWIR bands in the retrieval slightly increases the retrieval uncertainties for $R_{rs}$ at 410nm from 0.00031 to 0.00041.

### 4.4   NAAMES-Coastal waters (11/04/2015)

On November 4 during the NAAMES01 campaign in 2015, the aircraft flew over the Delaware Bay where there are strong

water leaving signals and small aerosol loadings. Here, the choice of the bio-optical model is more important than for the SABOR-Coastal case, where the water leaving signal is small. A location inside Delaware Bay is chosen to discuss the retrieval uncertainties and the impact of the bio-optical models as shown in Figure 1. Then, the retrieval over the whole flight track across Delaware bay is conducted and compared with MODIS ocean color product. The RSP measurement was made at noon with





the solar zenith angle near $60°$ as shown in Figure 2 (d) and in Table 1 for NAAMES-Coastal. The principal plane is almost perpendicular to the RSP scanning direction with a relative azimuth of $75°$. There is less influence from the sunglint as shown in Figure 13. No RSP SWIR bands are available for this dataset.

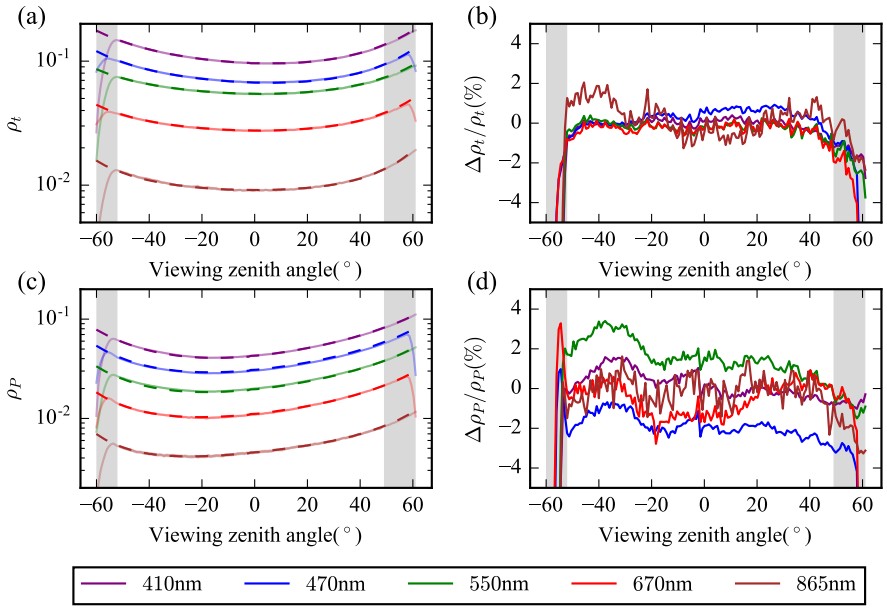

**Figure 13.** Same as Figure 3 but for NAAMES-Coastal on 11/04/2015. The minimum cost function value is $\chi^2_{min} = 0.16$ and the bio-optical model used here is Bio-2.

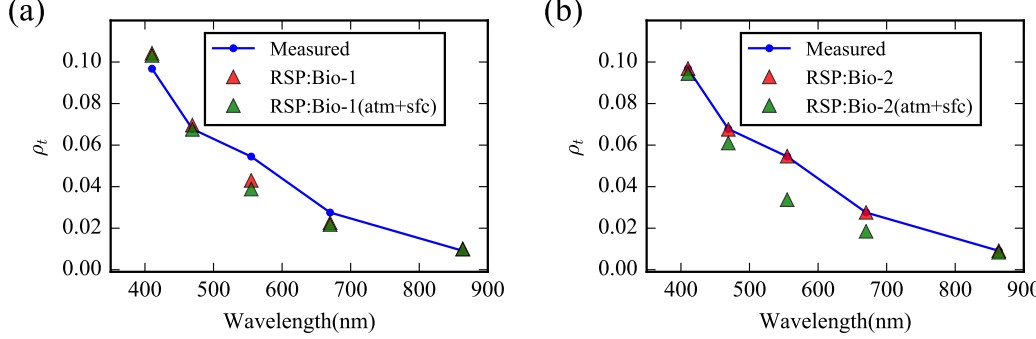

**Figure 14.** The Comparison of the measurement and the simulated total reflectance $\rho_t$ of the whole atmosphere and ocean system, and the total reflectance from only the atmosphere and ocean surface (atm+sfc): (**a**) The RSP retrieval using Bio-1, (**b**) The RSP retrieval using Bio-2.

The maximum cost function value is $\chi^2_{max} = 1.8$ with Bio-2, but increases to $\chi^2_{max} = 25.2$ with Bio-1. The larger cost

5  function value indicates a larger bias in the simulation. As we have discussed, Bio-1 works better for open water, as well as





some coastal water cases when the water leaving signal is small, such as SABOR-Coastal case. As shown in Figure 14 (a), for the retrieval using the Bio-1 model, the best fitted simulation results have a large cost function of $\chi^2_{min} = 19.6$. The simulated reflectance tends to overestimate the reflectance at shorter wavelength such as 410nm and underestimate the reflectance at longer wavelengths at 550nm and 670nm. This will results in a larger aerosol optical depth as shown in Figure 15(a) and a

negative remote sensing reflectance as shown in Figure 15(b). The reflectance with only the atmosphere and ocean surface (no ocean water body, denoted as "atm+sfc") is also shown in Figures 13. The difference between the total and the atm+sfc would be the contribution from the ocean water body only. When using the Bio-2 model, the comparison of the measured and best fitted simulation of $\rho_t$ is shown in Figure 14(b). A good agreement can be found with difference less than 1% at the nadir direction. The percentage difference for $\rho_t$ over the whole viewing direction used in retrieval is less than 2% in $\rho_t$ and less than

4% for $\rho_P$ as shown in Figure 13 (b) and (d). This results in a smaller aerosol optical depth as shown in Figure 15(a) and a reasonable remote sensing reflectance spectrum as shown in Figure 15(b).

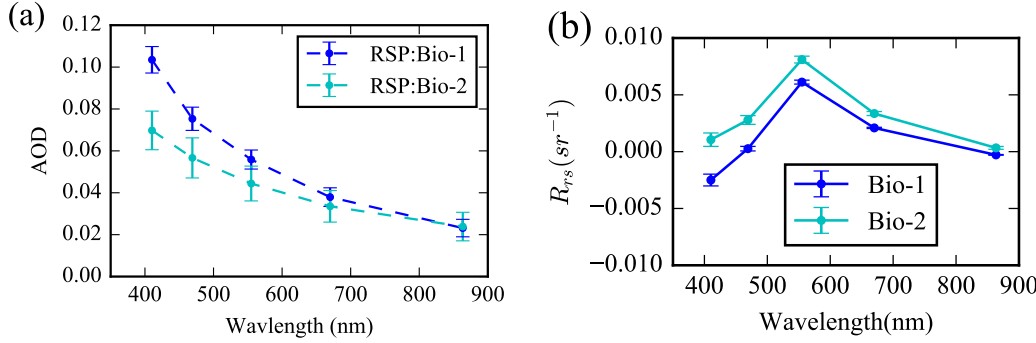

**Figure 15.** (**a**) The comparison of the retrieved AOD using the two bio-optical models: Bio-1 and Bio-2,(**b**) same as (**a**) but for the remote sensing reflectance. The error bars indicate the retrieval uncertainties.

In this case, the maximum remote sensing reflectance is almost three times of the maximum reflectance from the SABOR-Coastal case, thus requiring different consideration of the ocean signal through the bio-optical models in order to accurately conduct the joint retrieval algorithm for atmospheric correction. The retrieved optical depth and remote sensing reflectance strongly depend on the choice of the bio-optical models. When using Bio-1 and Bio-2, the retrieved AOD at 550nm is $0.056 \pm$

$0.005$ and $0.044+\pm0.008$ respectively. Using Bio-1 results in a smaller variability in the AOD retrieval, but much larger optical depths and negative remote sensing reflectances at shorter wavelength, suggesting that Bio-2 is necessary for this case. We use the averaged aerosol properties for the NAAMES-Coastal case as the initial values and conduct the joint retrieval along the whole flight track across the bay, the overall cost function is within $\chi^2 = 1.35$ which indicates good convergence along the

track.




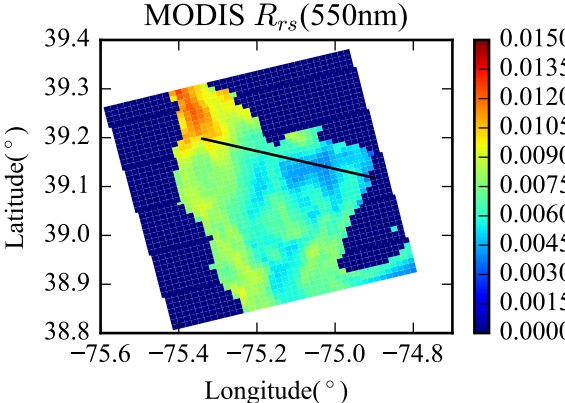

**Figure 16.** The remote sensing reflectance for 550nm band from the MODIS Ocean Color product. The black line indicates the RSP flight track.

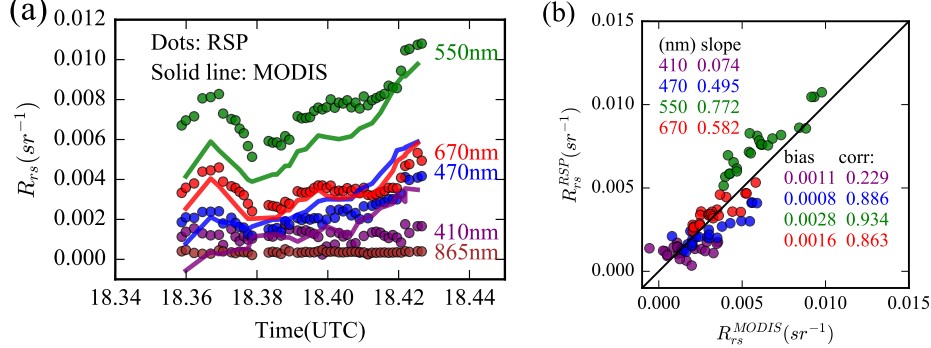

**Figure 17.** (a)The comparison of the RSP retrieved remote sensing reflectance across the Delaware Bay where dots indicate RSP retrieval and solid lines indicate the MODIS product. The time axis is from RSP measurement. (**b**) The correlation between the RSP and MODIS results with linear regression bias and slope shown for each wavelength.

The RSP retrieved $R_{rs}$ is compared with retrievals from MOIDS/Aqua over the Delaware Bay. The MODIS ocean color product was generated by the SeaDAS l2gen software, which includes the atmospheric correction algorithm proposed by Gordon and Wang (1994) that is more recently described in its algorithm theoretical basis document (NASA Ocean Color Web). The MODIS ocean color product provides $R_{rs}$ at 10 wavelength: 412, 443, 469, 488, 531, 547, 555, 645, 667, and 678nm respectively. Figure 16 shows the RSP track in the MODIS $R_{rs}$ image. The RSP pixels are collocated with the MODIS pixels within a distance of 500 meters. The MODIS bands of 412, 469, 555, and 667nm are chosen to compare the corresponding RSP bands of 410, 470, 550, and 670nm bands. The $R_{rs}$ from RSP and MODIS shows similar spatial variations in Figure 17 (a). Figure 17 (b) shows the correlation (corr) for each band with the linear regression slope and bias. The bands 470, 550 and 670 all show high correlation of 0.88, 0.93 and 0.86 respectively. The $R_{rs}$ from RSP and MOIDS agrees well for 550nm



across the whole track. We found RSP retrieved $R_{rs}$ are larger than the MOIDS retrievals with a value between 0.001 and 0.002 from 18.36 to 18.41 at 550nm, while smaller than the MODIS retrievals within a value within 0.001 from 18.40 to 18.43 at 410nm. In average, the mean absolute errors (MAE) between the RSP and MODIS retrievals are 0.0009, 0.0009, 0.0015 and 0.0005 for bands of 410,470,550,670nm; while the corresponding root mean square errors (RMSE) are 0.0011, 0.0011, 0.0016

and 0.0006. The possible reason for the discrepancy may be due to the different aerosol model retrieved/selected for RSP and MODIS using two completely different algorithms and datasets. MODIS relies on two NIR bands of 748nm and 869nm to determine the aerosol model, while RSP retrieval use all VIS and NIR bands from 410nm to 865nm and conduct retrieval on a coupled atmosphere and ocean system.

## 5   Discussions

The uncertainties of the remote sensing reflectance retrievals associated with different initial values in the optimization are evaluated and summarized in Table 5 for wavelengths from 410nm to 865nm. This uncertainty is due to the local minima of the cost function in the retrieval, and have not been quantified before for the study of atmospheric correction. Due to the large number of retrieval parameters and the non-linearity of the cost functions, the choice of the initial values often becomes important, and it is essential to understand the corresponding uncertainty and also its relationship with the PACE requirement

on atmospheric correction.

The PACE requirement on the atmospheric correction is to retrieve the normalized water leaving reflectance $[\rho_w(\lambda)]_N$ with an accuracy of the maximum of either $5\%$ or 0.001 over the spectral range of 400-710nm for open ocean conditions and standard marine atmosphere (PACE, 2012). Since the normalized water leaving reflectance can be related to the remote sensing reflectance through $[\rho_w(\lambda)]_N = \pi R_{rs}$ (Mobley et al., 2016), the PACE atmospheric requirement would be the maximum of

either $5\%$ or 0.0003 in $R_{rs}$. The computed requirement accuracy for PACE is listed in the Table 5 in comparing with the RSP retrieval accuracy.

For the open water cases, the retrieval uncertainty for the remote sensing reflectance is smaller than the PACE atmospheric correction requirement for all the bands except for 410nm. In SABOR-Open case, the retrieval uncertainty at 410nm is 0.0008 which is larger than the PACE requirement of 0.0005. Similarly for NAAMES-Open, the retrieval uncertainty for Rrs at 410nm

is 0.00031 which is slightly larger than the PACE requirement of 0.0003.

For coastal waters, it is more challenging to retrieve the remote sensing reflectance accurately due to the complex water properties. Since the PACE atmospheric correction for coastal waters is not available, we use the same PACE requirement for open water in comparison. Both the two bio-optical models are applied in the coastal water cases, for NAAMES-Coastal case, only Bio-2 provided reasonable result, while Bio-1 results in negative value of $R_{rs}$ at shorter wavelengths. The retrieval

uncertainties at 410nm and 470nm with values of 0.00059 and 0.00039 are larger than the PACE requirement of 0.0003 for both bands as shown in Table 4. All the other bands are well within the requirement. In this coastal water case, the water leaving signal is strong, and it is therefore important to select the bio-optical model to provide proper constraints of the water leaving signal in the coupled atmosphere and ocean system.





For SABOR-Coastal case, due to the large aerosol loading and small water leaving signals, both bio-optical models demonstrated comparable results in aerosol and water leaving signal results. Due to the larger number of retrieval parameters for Bio-2, the retrieval uncertainties are larger, with a maximum number of 0.001, 0.00066 and 0.00049 at 410nm, 470nm and 550nm, respectively, which are all larger than the PACE requirement of 0.0003. When using Bio-1, the retrieval uncertainties

are much reduced with only the uncertainty at band 410nm larger than the PACE requirement. Meanwhile, both bio-optical models result in high accuracy in the retrieval of the AOD as compared with the AERONET and HSRL AOD product. Furthermore, two treatment of the refractive index spectra are compared in the retrieval for SABOR-Coastal case. When using the PCA representation of aerosol refractive indices, there is a dip at 470 nm in the spectra shape of the retrieved remote sensing reflectance, which is different from the in situ measurement. This suggest that the aerosol refractive index spectrum may have

small spectral variation which is not captured by the smooth representation of PCA. After introducing a small adjustment of the refractive index at the band of 410 and 470nm in the retrieval, the retrieved remote sensing reflectance resemble similar shape with the in situ measurement. Both treatments of the refractive index carry similar uncertainties. If additional collocated datasets are available for validation in the future, we will further investigate and attempt to identify the best representation of the refractive index spectrum beyond PCA with a certain amount of flexibility and stability.

The joint retrieval algorithm can provide acurate retrieval of the aerosol properties as compared with the HSRL and AERONET AOD over both open and coastal waters. The remote sensing reflectance can also be accurately retrieved as compared with the in situ measurements. Meanwhile, the measurement dataset needs to be carefully examined to remove all the possible influence from cloud and other error sources. The retrieval with the measurement over sunglint also requires close examination where the sunglint may be influenced by wind direction or instantaneous ocean surface slopes with large waves that are not described

in the forward model. Overall the retrieval uncertainties are comparable with the PACE atmospheric correction requirement, but a higher uncertainties are always associated with the deep blue band of 410nm.

The retrieval uncertainties associated with the local minimum are discussed, which can help to determine better initial values and quantify the accuracy. The uncertainties from the error propagation of the instrument noise can also be evaluated with selected initial value and provide another aspect of the retrieval uncertainties (Rogers, 2000). The joint retrieal algorithm

will be applied to HARP2 and SPEXone to evaluate the possible accuracy directly relevant to the PACE mission.



**Table 5.** The atmospheric correction uncertainties for the four cases as listed in Table 1 for the RSP retrieval. The uncertainties are computed through using different initial values in the optimization. The PACE requirement (values in the parenthesis) is also shown. The numbers in bold indicate the RSP retrieval uncertainties larger than PACE requirement. Bio-1 is the [Chla]-based bio-optical model used for open waters, and Bio-2 is the generalized bio-optical model used for coastal waters. For the SABOR-Coastal case where the aerosol loading is larger and the water leaving signal is small; both bio-optical models are computed for discussion. All cases use the seven RSP bands except for the one indicated by asterisk which did not use SWIR bands.

| Case | Bio1/Bio2 | AOD(550) | $\Delta R_{rs}(410)$ | $\Delta R_{rs}(470)$ | $\Delta R_{rs}(550)$ | $\Delta R_{rs}(670)$ | $\Delta R_{rs}(865)$ |
|---|---|---|---|---|---|---|---|
| SABOR-Open | Bio-1 | 0.17 | **0.00080** | 0.00031 | 0.00008 | 0.00013 | 0.00006 |
| | | | (0.00058) | (0.00031) | (0.0003) | (0.0003) | (0.0003) |
| SABOR-Coastal | Bio-2 | 0.34 | **0.00100** | **0.00066** | **0.00049** | 0.00029 | 0.00013 |
| | Bio-1 | | **0.00035** | 0.00018 | 0.00008 | 0.00010 | 0.00006 |
| | | | (0.0003) | (0.0003) | (0.0003) | (0.0003) | (0.0003) |
| NAAMES-Open | Bio-1 | 0.14 | **0.00031** | 0.00020 | 0.00002 | 0.00001 | 0.00000 |
| | | | (0.0003) | (0.0003) | (0.0003) | (0.0003) | (0.0003) |
| NAAMES-Coastal | Bio-2* | 0.06 | **0.00059** | **0.00039** | 0.00030 | 0.00018 | 0.00012 |
| | | | (0.0003) | (0.0003) | (0.00041) | (0.0003) | (0.0003) |

## 6 Conclusions

We have developed a joint retrieval algorithm for aerosol and water leaving properties based on a radiative transfer model for coupled atmosphere and ocean systems. Both the aerosol optical properties and ocean bio-optical properties are flexible in order to model complex coastal scenes. The algorithm has been validated for synthetic measurements in a previous study. In this study, we applied the joint retrieval algorithm to RSP airborne measurements. Four cases from SABOR and NAAMES field campaigns are chosen with two open and two coastal water cases. Our retrieval results indicate a good agreement in the aerosol optical depth compared with both the HSRL and AERONET products, and also a good agreement of the remote sensing reflectance as compared with in situ measurements and the MODIS ocean color products.

Two different but related bio-optical models are implemented and discussed in the joint retrieval algorithm for the study of atmospheric correction over different water conditions. For open waters, the [Chla]-based bio-optical model (Bio-1) is used with a single parameter to define all seawater components, while for coastal waters, a seven parameter bio-optical model (Bio-2) is employed. To understand the applicability of the two bio-optical models, both models are applied in the coastal waters cases from SABOR and NAAMES. For the SABOR coastal water cases, the water leaving signal is weak and both bio-optical models provide similar results of aerosol and the remote sensing reflectance retrieval, but there is smaller uncertainty associated with Bio-1. For the NAAMES coastal waters, the water leaving signal is relatively strong and only Bio-2 can provide a reasonable remote sensing reflectance retrieval that avoids negative values.



Using RSP retrievals in open waters as a proxy, we show that this joint retrieval can nearly meet PACE mission requirements for atmospheric correction at shorter wavelengths (e.g., 410nm) and performs well within the requirement at longer wavelengths. For coastal waters, the appropriate bio-optical model may be selected depending on the magnitude of the water leaving signal and the uncertainty requirement. Generally, Bio-2 may have larger uncertainties compared with Bio-1 due to its

larger parameter space, but it is necessary to use Bio-2 in order to better fit the data and avoid negative reflectance retrievals for certain cases. A comparison with the MODIS ocean color product shows high correlation but also differences in magnitudes in remote sensing reflectances.

The cases we studied cover various aerosol loadings, viewing geometry and sunglint conditions providing a useful quantification for the retrieval uncertainties of both aerosol and water leaving signals in the study of atmospheric correction using the

multi-angle, wavelength and polarization measurements. It provides useful understanding to better harvest the rich information in such measurements, and to reduce the possible influence from various error sources such as clouds and sunglint. The algorithm provides a flexible description of the aerosol and ocean bio-optical properties, when combined with more co-located remote sensing reflectances and in situ measurement, a more efficient algorithm may be developed to reduce and optimize the retrieval parameters in the algorithm. The lessons discussed and the accuracy evaluated from the retrieval with the polarization

measurement for atmospheric correction can assist the future development of the atmospheric correction algorithm for the PACE mission, with the goal of combining both the OCI and polarimeter measurements.

## Appendix A: RSP File list for processing

As shown in Table 1 and Figure 1, four cases from the RSP measurements are studied from both the SABOR and NAAMES campaigns. The corresponding RSP L1B data can be located from NASA GISS website (NASA RSP Data Site). The file names

are listed as follows

- SABOR-open (07/27/2014):
  RSP1-UC12_L1B-RSPGEOL1B-GeolocatedRadiances_20140727T141100Z_V001-20160518T201607Z.h5

- SABOR-Coastal (07/30/2014):
  RSP1-UC12_L1B-RSPGEOL1B-GeolocatedRadiances_20140730T151114Z_V001-20160518T213810Z.h5

- NAAMES-open (05/26/2016):
  RSP1-C130_L1B-RSPGEOL1B-GeolocatedRadiances_20160526T150519Z_V001-20160601T174243Z.h5

- NAAMES-coastal (11/04/2015):
  RSP1-C130_L1B-RSPGEOL1B-GeolocatedRadiances_20151104T182047Z_V002-20161129T190435Z.h5





## Appendix B:  Remote sensing reflectance representation

The remote sensing reflectance defined in this study is represented in Eq. 2 as:

$$R_{rs} = \frac{\rho_w^{Sensor}}{\pi t_d t_u} \tag{B1}$$

where $t_d$ is the downward transmittance of the solar irradiance to the surface, and $t_u$ is the upward transmittance of the water

leaving radiance to the detector. To compute the remote sensing reflectance quantitatively, $\rho_w^{Sensor}$ can be obtained from the difference between the measurement $\rho_t$ and the simulated reflectance $\rho_{t,atm+sfc}^{f,Sensor}$ at the sensor considering only the atmosphere and ocean surface (denoted by "atm+sfc" following the notation by Bo-Cai Gao (Gao et al., 2000)):

$$\rho_w^{Sensor} = \rho_t - \rho_{t,atm+sfc}^{f,Sensor} \tag{B2}$$

Both $t_d$ and $t_u$ are due to the scattering and absorption in the atmosphere and can be computed from the radiative transfer

simulation. Specifically, the transmittance $t_d$ is defined as the ratio of downwelling irradiance $F_d^{f,+}$ just above the ocean surface with respect to the solar irradiance $F_0$ as:

$$t_d = \frac{F_d^{f,+}}{\mu_0 F_0}, \tag{B3}$$

where $F_d^{f,+}$ is computed from the forward model using the retrieved atmosphere properties. $t_u$ is the transmittance of the upwelling water leaving radiance from surface to the sensor at certain viewing direction $\theta_v$, which can be estimated as

$$t_u(\theta_v) = \frac{L_t^{f,Sensor}(\theta_v) - L_{t,atm+sfc}^{f,Sensor}(\theta_v)}{L_t^{f,+}(\theta_v) - L_{t,atm+sfc}^{f,+}(\theta_v)}, \tag{B4}$$

where all the quantities are computed from the forward model, $L_t^{f,Sensor}$ represents the total radiance at sensor and $L_t^{f,+}$ represents the radiance just above ocean surface computed from the forward model with the total atmosphere and ocean system; same for $L_{t,atm+sfc}^{f,Sensor}$ and $L_{t,atm+sfc}^{f,+}$ but considered only the atmosphere and ocean surface with no scattering in the ocean. Therefore $L_t^{f,+}(\theta_v) - L_{t,atm+sfc}^{f,+}(\theta_v)$ represents the water leaving radiance just above the ocean surface, and $L_t^{f,Sensor}(\theta_v) -$

$L_{t,atm+sfc}^{f,Sensor}(\theta_v)$ represents the water leaving radiance which transmits to the the sensor. Note that since the water leaving reflectance is generally small, we ignored the contribution from the reflection of the water leaving signals by the atmosphere back to the ocean.

## Appendix C:  RSP noise model

The RSP uncertainty model used in this study is summarized as follows. More details are in Knobelspiesse et al. (2019). The

error covariance $\sigma_{\rho_t}$ and $\sigma_{\rho_P}$ for radiance and polarized radiance in Eq. 3 are defined as the sum of noise and calibration





uncertainties:

$$\sigma_{\rho_t}^2 = \sigma_{\rho_t}^2(noise) + \sigma_{\rho_t}^2(calibration) \tag{C1}$$

$$\sigma_{\rho_t}^2(noise) = \left(\frac{r^2\sigma'_{floor}}{\mu_0}\right)^2 + \frac{r^2 a' \rho_t}{2\mu_0} \tag{C2}$$

$$\sigma_{\rho_t}^2(calibration) = \frac{\sigma_{\ln K}^2}{16}\rho_P^2 + \sigma_{\alpha_c}^2 \rho_t^2 \tag{C3}$$

Same for the total uncertainty of polarized reflectance uncertainty model:

$$\sigma_{\rho_P}^2 = \sigma_{\rho_P}^2(noise) + \sigma_{\rho_P}^2(calibration) \tag{C4}$$

$$\sigma_{\rho_P}^2(noise) = 4\left(\frac{r^2\sigma'_{floor}}{\mu_0}\right)^2 + 2\frac{r^2 a' \rho_t}{\mu_0} \tag{C5}$$

$$\sigma_{\rho_P}^2(calibration) = \frac{\sigma_{\ln K}^2}{2}\rho_t^2 + (\sigma_{\ln\alpha}^2 + \sigma_{\alpha_c}^2)\rho_P^2 \tag{C6}$$

Two RSP instruments has been built with name RSP1 and RSP2. In our study, the measurement are only from RSP1 with

noises and uncertainties including detector floor noise $\sigma'_{floor} = 2.0 \times 10^{-5}$, shot noise parameter $a' = 1.0 \times 10^{-7}$, relative gain coefficient characterization uncertainty $\sigma_{\ln K} = 0.005$, absolute radiometric characterization uncertainty $\sigma_{\alpha_c} = 0.015$, and polarimetric characterization uncertainty $\sigma_{\ln\alpha} = 0.002$. Solar distance ($r$) is in astronomical units with a value of 1.0. RSP2 has slightly different noise model which is not discussed here.

## Appendix D:  Bio-optical model for open waters

The [Chla]-based bio-optical model (Bio-1) can be derived from the generalized bio-optical model (Bio-2) by imposing constraints on its parameters for the study of open waters (Zhai et al., 2015, 2017). The phytoplankton absorption coefficients $a_{ph}$ is the same as in Table 3. Since no contribution from NAP is assumed for open waters, the particulate absorption coefficients $a_{dg}$ depends only on phytoplankton. Its parameter $a_{dg}(440)$ is specified by [Chla] as:

$$a_{dg}(440) = p_2 \cdot a_p(440, [Chla]) \tag{D1}$$

$$p_2 = 0.3 + \frac{5.7 \cdot R_2 \cdot a_p(440, [Chla])}{0.02 + a_p(440, [Chla]} \tag{D2}$$

where $R_2 = 0.5$ is assumed and a fixed value of $S_{dg} = 0.018$ is used.

The particulate scattering coefficients $b_{bp}$ also depends only on phytoplankton for open water studies. Its parameter $b_{bp}(660)$ is specified by [Chla] as

$$b_{bp}(660) = 0.347[Chla]^{0.766} \tag{D3}$$

The spectral slope is specified as $S_{bp} = -0.5(\log_{10}[Chla]-0.3)$ for $0.02 < [Chla] < 2mg/m^3$, otherwise zero. The particulate backscattering fraction can be specified as $B_p = 0.002 + 0.01(0.5 - 0.25\log_{10}[Chla])$ where $B_p$ is assumed to be spectrally flat (Huot et al., 2008).



*Author contributions.* M.G. and P.-W.Z. developed the retrieval algorithm and generated the scientific data used in this paper. M.G. wrote the original manuscript. P.-W Z. and B.F. formulated the original concept for this study. K. K. advised on the RSP noise model. B.C. advised on the use of RSP data and the retrieval sensitivity of aerosols. P.J.W, A.I. and Y.H. advised on the ocean bio-optical models. A.C. provided and analyzed the HyperPro measurement data. All authors participated the writing and editing of this paper.

5   *Competing interests.* The authors declare no conflict of interest.

*Acknowledgements.* This research was funded by NASA Grants (80NSSC18K0345 and NNX15AK87G). Y. Hu was funded by the NASA Radiation Science program administered by Hal Maring and the Biology and Biogeochemistry program administered by Paula Bontempi. The authors would like to thank the NAAMES and SABOR teams, including the ship's crew and captains of R/V Endeavor and R/V Atlantis; the NASA AERONET team and the NASA Langley team for maintaining the AERONET COVE_SEAPRISM site.

10   The hardware used in the computational studies is part of the UMBC High Performance Computing Facility (HPCF). The facility is supported by the U.S. National Science Foundation through the MRI program (grant nos. CNS–0821258, CNS–1228778, and OAC–1726023) and the SCREMS program (grant no. DMS–0821311), with additional substantial support from the University of Maryland, Baltimore County (UMBC). See hpcf.umbc.edu for more information on HPCF and the projects using its resources.



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
