# Peer review of "Inversion of multi-angular polarimetric measurements over open and coastal ocean waters: a joint retrieval algorithm for aerosol and water leaving radiance properties"

_Atmospheric Measurement Techniques, 2019_

## Referee Comment (RC1) · Anonymous Referee #2 · 23 May 2019

The manuscript by Gao et al., "Inversion of multi-angular polarimetric measurements over open and coastal ocean waters: a joint retrieval algorithm for aerosol and water leaving radiance properties" presents a study to apply the joint retrieval algorithm (to obtain the aerosol and water leaving signal simultaneously) to RSP airborne measurements. This retrieval algorithm has been validated with synthetic data earlier, while in this study the focus was to evaluate it against airborne polarimetric measurements from the Research Scanning Polarimeter (RSP) over both open and coastal ocean waters acquired in two field campaigns: the Ship-Aircraft Bio-Optical Research (SABOR)

[Figure]

in 2014 and the North Atlantic Aerosols and Marine Ecosystems Study (NAAMES) in 2015 and 2016. Thus the focus of the paper is clearly defined and targeted, and so are the results and conclusions that are presented. I think the manuscript suits the scope of AMT and deserves to be published. I have only few relatively minor comments that I wish are considered in the revised version.

The main comment has to do with aerosol "treatment" of the algorithm, for which I thought some further discussion might be suitable. For instance, regarding the "perturbations to the real and imaginary parts of the PCA refractive indices at 410nm and 470nm". I was thinking that it was likely the allowed perturbation particularly in imaginary index that resulted in the improvement of Figure 10b. However, this was not discussed, so the question remained which one more effectively influenced the results by these "perturbations"? If it was imaginary index, then this it is likely related to the spectral dependence of absorption by organic aerosols (Brown Carbon). It is well known that in this regard, what we nowadays know about spectral aerosol absorption, the Shettle and Fenn 1979 does not represent this understanding well. Perhaps these issues could be discussed in the revised manuscript.

About the Figures 4,8: is it so that you do not show unitless AOD? I thought it should be extinction in 1/km, but it seems it is something else in the unit,since from that Figure I would estimate much larger AOD than what is shown for HSRL in the Figure 9b. So good to clarify what exactly is shown by these type of figures.

Page 14, line #4. You mention that the retrieval produces larger aerosol absorption. What does this mean exactly and how it was concluded (comparing against AERONET AAOD)? This is not clear, since it seems that you retrieve only aerosol extinction and single scattering albedo is based on your assumed aerosol model. Please clarify this statement.

- In many plots Wavlength -> Wavelength

- Page 2, line #29: soley -> solely

---

## Referee Comment (RC2) · Anonymous Referee #1 · 24 May 2019

The paper by Gao et al describes the application of a joint retrieval algorithm for aerosol properties and water leaving radiances (WLR) to multi-angle measurements of radiance and polarization from RSP for different situations (open ocean, coastal waters, low and high aerosol load). They compare 2 different bio-optical models and find that a more complex model with 7 parameters is needed for coastal waters if the WLR is high (and aerosol load low) while for open ocean or coastal waters with low WLR a simple model just depending on the CHL - a concentration is sufficient, or even better. The paper is very relevant to the NASA PACE mission. I recommend publication after

addressing my comments below.

**General Comments**

- The description of the inversion method needs to be somewhat extended. It is mentioned that the cost function of Eq.3 is being minimized. I am surprised that there is no regularization term in the form of a side constraint (i.e. difference with prior or smoothness) included in the cost function. It might be that the authors implicitly include regularization through the Levenberg-Marquardt (LM) method, because in this method the difference with the previous iteration step is being minimized. If this is the case, it should be explicitly mentioned that regularization is brought in through the LM method. Although this is common practice, it is a non-optimal way of including regularization (see e.g. Rodgers, 2000).

- The approach of uncertainty estimation through an ensemble approach with different 1st guess state vector is very interesting and provides useful insight in the retrieval result. However, I find that the resulting uncertainties are over-interpreted when it comes to trading these uncertainties against the PACE requirements. As the authors note themselves in the paper, it can happen that a retrieval with a wrong model leads to a smaller uncertainty but the retrieval result is obviously worse (i.e. due to a bias) than the retrieval with a more correct model but a larger uncertainty (found from the ensemble approach). So, I suggest to remove this discussion from the paper or at the very least provide the right perspective. Something that could be compared against the PACE requirement is the difference between a retrieval result and a validation measurement, although also here one has to be very careful given the small sample. - For the case with high aerosol load, the authors adjust the imaginary (part of the) refractive index (IRI) in a rather ad hoc way by changing the value at 410 and 470 nm. It seems that the spectra from 'd Almeida do not include the right spectral variation for all aerosol types. I would advise the authors to see how things change if they also include Brown Carbon in the PCA analysis, using the IRI spectra of Kirchstetter, et al., (2004), (Evidence that the spectral dependence of light absorption by aerosols is affected by organic carbon,

[Figure]

J. Geophys. Res., 109, D21208, doi:10.1029/2004JD004999.) At least this possible
solution should be discussed in the paper.

**Minor comments**: - p2, l25: Correct reference for SPEXone is: Hasekamp et al.,
JQSRT, 227, 170 - 184, 2019, doi: https://doi.org/10.1016/j.jqsrt.2019.02.006. The correct reference for the underlying polarimetric measurement technique (spectral modulation) is: Snik et al, Appl. Opt., 48(7):1337-46, 2009, doi:10.1364/AO.48.001337.

- p3, l23: typo "measurments"

- p3, l31: It would be useful to include for the different cases investigated in the paper
an indication of the error on radiance and polarized radiance that result from the model.

- p9, l16: For cloud screening based on goodness-of-fit, please refer to Stap et al.,
(2015). Sensitivity of parasol multi-angle photo-polarimetric aerosol retrievals to cloud
contamination. Atmospheric Measurement Techniques, 8 (3), 12871301. Retrieved
from https://www.atmos-meas-tech.net/8/1287/2015/ doi: 10.5194/amt-8-1287-2015

- p12, l5: The difference between RSP and HSRL=0.015. This seems well within the
1-sigma error so why do you expect it is caused by the different viewing geometry? At
least mention that the difference is within 1-sigma error.

- p14, l1: "relative viewing azimuth" –> "relative azimuth"

- p14, l9: It seems chi2 is larger for the model with more parameters while I would
expect better capability to fit the measurement with more parameters. Please explain.

- p18, l10-11: "The maximum uncertainties for AOD are at 410nm with a value of
0.009". How does this relate to the error in AOD of 0.017 quoted one sentence earlier?

- p22, l1: "MOIDS" –> "MODIS"

---

## Author Comment (AC1) · 6 Jun 2019

Dear reviewer,
We appreciate the constructive comments, which are very helpful to improve the clarity of the manuscript. We have addressed every point in the revised manuscript, which are detailed below in blue.

Anonymous Referee #2
The manuscript by Gao et al., "Inversion of multi-angular polarimetric measurements over open and coastal ocean waters: a joint retrieval algorithm for aerosol and water leaving radiance properties" presents a study to apply the joint retrieval algorithm (to obtain the aerosol and water leaving signal simultaneously) to RSP airborne measurements. This retrieval algorithm has been validated with synthetic data earlier, while in this study the focus was to evaluate it against airborne polarimetric measurements
from the Research Scanning Polarimeter (RSP) over both open and coastal ocean waters acquired in two field campaigns: the Ship-Aircraft Bio-Optical Research (SABOR) in 2014 and the North Atlantic Aerosols and Marine Ecosystems Study (NAAMES) in 2015 and 2016. Thus the focus of the paper is clearly defined and targeted, and so are the results and conclusions that are presented. I think the manuscript suits the scope of AMT and deserves to be published. I have only few relatively minor comments that I wish are considered in the revised version.

The main comment has to do with aerosol "treatment" of the algorithm, for which I thought some further discussion might be suitable. For instance, regarding the "perturbations to the real and imaginary parts of the PCA refractive indices at 410nm and 470nm". I was thinking that it was likely the allowed perturbation particularly in imaginary index that resulted in the improvement of Figure 10b. However, this was not discussed, so the question remained which one more effectively influenced the results by these "perturbations"? If it was imaginary index, then this it is likely related to the spectral dependence of absorption by organic aerosols (Brown Carbon). It is well known that in this regard, what we nowadays know about spectral aerosol absorption, the Shettle and Fenn 1979 does not represent this understanding well. Perhaps these issues could be discussed in the revised manuscript.

We agree that the PCA representation of aerosol refractive index based on Shettle and Fenn (1979) may not be sufficient for all cases encountered. We further revised the paragraph as follows:
"Furthermore, there may be small variations in the aerosol refractive index spectrum that are not captured by the smooth representation of the PCA, which may affect the retrieval of water leaving radiance adversely. For example, organic carbon may introduce spectral dependency of light absorption (Kirchstetter, 2004), but is not considered in the datasets used for the PCA computation."

Regarding which part, real or imaginary, of the refractive index affects the results more, our observation is that both play a role in the fitting. In our revised manuscript, we added:
"…A better agreement of the spectral shape of the retrieved Rrs can be found for both bio-optical models as shown in Figure 10(b), which is due to the additional refractive index spectral perturbation. The retrieved aerosol volume density is dominated by the fine mode aerosols with the mean values of the real refractive indices of 1.58, 1.55, 1.51 at 410, 470, 550nm, which deviates from the PCA representation by 0.06, 0.04, and 0.003.  Meanwhile, the mean values for

the imaginary refractive indices are 0.014,0.021, 0.011 at 410, 470, 550nm, which differ from the PCA representation by 0.006, 0.014, and 0.004 ."

About the Figures 4,8: is it so that you do not show unitless AOD? I thought it should be extinction in 1/km, but it seems it is something else in the unit,since from that Figure I would estimate much larger AOD than what is shown for HSRL in the Figure 9b. So good to clarify what exactly is shown by these type of figures.
The cumulative AOD shown in Figures 4 and 8 are unitless. They are the AOD of the layer from the aircraft to the altitude shown in the plots.  The caption of the Figure 4 is revised accordingly: "The cumulative aerosol optical depth (AOD) from HSRL, which is the AOD of the layer from the aircraft to the altitude as indicated in the plot."

Page 14, line #4. You mention that the retrieval produces larger aerosol absorption. What does this mean exactly and how it was concluded (comparing against AERONET AAOD)? This is not clear, since it seems that you retrieve only aerosol extinction and single scattering albedo is based on your assumed aerosol model. Please clarify this statement.
We do not assume an aerosol model. The volume concentrations of six modes are retrieved. Also the spectral refractive index for both fine and large modes are retrieved. Based on this information, the aerosol single scattering albedo can be calculated. To make this clearer, we added a sentence in the section 3.2 (page 7): "The PCA coefficients for both the real and imaginary refractive indices are retrieved from the algorithm."

- In many plots Wavlength -> Wavelength
Corrected in the revised manuscript.

- Page 2, line #29: soley -> solely
Corrected in the revised manuscript.

---

## Author Comment (AC2) · 6 Jun 2019

Dear reviewer,
We really appreciate your constructive comments and suggestions, which are very helpful to improve the clarity of the manuscript. We have addressed every point in the revised manuscript, which are detailed below in blue:

Anonymous Referee #1
The paper by Gao et al describes the application of a joint retrieval algorithm for aerosol properties and water leaving radiances (WLR) to multi-angle measurements of radiance and polarization from RSP for different situations (open ocean, coastal waters, low and high aerosol load). They compare 2 different bio-optical models and find that a more complex model with 7 parameters is needed for coastal waters if the WLR is high (and aerosol load low) while for open ocean or coastal waters with low WLR a simple model just depending on the CHL - a concentration is sufficient, or even better. The paper is very relevant to the NASA PACE mission. I recommend publication after addressing my comments below.

General Comments
- The description of the inversion method needs to be somewhat extended. It is mentioned that the cost function of Eq.3 is being minimized. I am surprised that there is no regularization term in the form of a side constraint (i.e. difference with prior or smoothness) included in the cost function. It might be that the authors implicitly include regularization through the Levenberg-Marquardt (LM) method, because in this method the difference with the previous iteration step is being minimized. If this is the case, it should be explicitly mentioned that regularization is brought in through the LM method. Although this is common practice, it is a non-optimal way of including regularization (see e.g. Rodgers, 2000).

We agree with the reviewer on the treatment of regularization. The LM method implies an implicit Twomey-Tikhonov regularization, which is introduced through a scaling matrix in the formulation of Moré (1978). The manuscript is revised as follows with two extra references added (Moré,1978; Rogers, 2000):
"The optimization algorithm used in this study is the Levenberg-Marquet method (Moré, 1980), where the Twomey-Tikhonov regularization is assumed implicitly (Moré,1978; Rogers,2000)".

- The approach of uncertainty estimation through an ensemble approach with different 1st guess state vector is very interesting and provides useful insight in the retrieval result. However, I find that the resulting uncertainties are over-interpreted when it comes to trading these uncertainties against the PACE requirements. As the authors note themselves in the paper, it can happen that a retrieval with a wrong model leads to a smaller uncertainty but the retrieval result is obviously worse (i.e. due to a bias) than the retrieval with a more correct model but a larger uncertainty (found from the ensemble approach). So, I suggest to remove this discussion from the paper or at the very least provide the right perspective. Something that could be compared against the PACE requirement is the difference between a retrieval result and a validation measurement, although also here one has to be very careful given the small sample.

Thank you for the interests and comments on the uncertainty evaluation. The uncertainties demonstrate how much influence on the retrieval results from different initial values and this is an important portion of the total retrieval uncertainties. The comparison with the PACE requirements provides useful guidance on the retrieval algorithm development where the influence of the choice of initial values cannot be avoided.

As the reviewer observed, in the manuscript we indeed demonstrated a case where a wrong ocean IOP model (SABOR-Coastal cases, bio-1 model) results a small uncertainty but a large bias (chi^2) as compared with bio-2 model. This is why it is important to evaluate both the bias (chi^2, table 4) and the uncertainty (table 5) in our discussion, and the wrong ocean IOP model with a large bias should be excluded for fair uncertainty comparison.

We revised the opening of section 5 to improve the clarity:
"The uncertainties of the remote sensing reflectance retrievals associated with different initial values in the optimization are evaluated and summarized in Table 5 for wavelengths from 410nm to 865nm, where the SABOR-Coastal case with the Bio-1 model is excluded due to its large bias in fitting the measurement as shown in Table 4."

- For the case with high aerosol load, the authors adjust the imaginary (part of the) refractive index
(IRI) in a rather ad hoc way by changing the value at 410 and 470 nm. It seems that the spectra from 'd Almeida do not include the right spectral variation for all aerosol types. I would advise the authors to see how things change if they also include Brown Carbon in the PCA analysis, using the IRI spectra of Kirchstetter, et al., (2004), (Evidence that the spectral dependence of light absorption by aerosols is affected by organic carbon, J. Geophys. Res., 109, D21208, doi:10.1029/2004JD004999.) At least this possible solution should be discussed in the paper.

Thank the reviewer for providing the reference on organic carbon. We addressed the lack of organic carbon in the PCA calculation as follows:
"…Furthermore, there may be small variations in the aerosol refractive index spectrum that are not captured by the smooth representation of the PCA, which may affect the retrieval of water leaving radiance adversely. For example, organic carbon may introduce spectral dependency of light absorption (Kirchstetter,2004), but is not considered in the datasets used for the PCA computation."

Furthermore, we added more details of the retrieved refractive indices:
"…A better agreement of the spectral shape of the retrieved Rrs can be found for both bio-optical models as shown in Figure 10(b), which is due to the additional refractive index spectral perturbation. The retrieved aerosol volume density is dominated by the fine mode aerosols with the mean values of the real refractive indices of 1.58, 1.55, 1.51 at 410, 470, 550nm, which deviates from the PCA representation by 0.06,0.04, and 0.003.  Meanwhile, the mean values for the imaginary refractive indices are 0.014,0.021, 0.011 at 410, 470, 550nm, which differ from the PCA representation by 0.006, 0.014, and 0.004."

The retrieved imaginary refractive index with perturbation is peaked at 470nm, which is different with the spectral shape of organic carbon, and requires more validation dataset to confirm its physical origin.

Minor comments: - p2, l25: Correct reference for SPEXone is: Hasekamp et al., JQSRT, 227, 170 - 184, 2019, doi: https://doi.org/10.1016/j.jqsrt.2019.02.006. The correct reference for the underlying polarimetric measurement technique (spectral modulation) is: Snik et al, Appl. Opt., 48(7):1337-46, 2009, doi:10.1364/AO.48.001337.
The reference for SPEXone is updated.

- p3, l23: typo "measurments"
Corrected in the revised manuscript.

- p3, l31: It would be useful to include for the different cases investigated in the paper an indication of the error on radiance and polarized radiance that result from the model.
We included an estimated modeling error in the cost function. This is mentioned in page 7, line 6 and restated below:
"The total uncertainty includes the instrument measurement uncertainties as discussed in Appendix C, the variance from averaging nearby RSP pixels (5 pixels are used in this study, which corresponds to a surface pixel size of approximately 500 meters), and the modeling uncertainties with an estimated percentage error similar to the measurement uncertainty."

- p9, l16: For cloud screening based on goodness-of-fit, please refer to Stap et al., (2015). Sensitivity of parasol multi-angle photo-polarimetric aerosol retrievals to cloud contamination. Atmospheric Measurement Techniques, 8 (3), 12871301. Retrieved from https://www.atmos-meas-tech.net/8/1287/2015/ doi: 10.5194/amt-8-1287-2015
The reference is added. Thank you for the suggestion.

- p12, l5: The difference between RSP and HSRL=0.015. This seems well within the 1-sigma error so why do you expect it is caused by the different viewing geometry? At least mention that the difference is within 1-sigma error.
Thank you for spotting the issue. The 1-sigma uncertainty for RSP retrieval is for 550nm in order to compare with HSRL AOD at 532nm, but in the manuscript it has been mistaken for a shorter wavelength. (This can be verified from Fig 5(a)). The corrected values are updated, and the paragraph is further revised as follows:
"…At UTC=14.231, the averaged RSP AOD at 550 nm is 0.15, which is larger than the HSRL AOD value (0.135). The difference is smaller than the 1-sigma uncertainty of RSP AOD retrieval, which is 0.017."

- p14, l1: "relative viewing azimuth" –> "relative azimuth"
Corrected in the revised manuscript.

- p14, l9: It seems chi2 is larger for the model with more parameters while I would expect better capability to fit the measurement with more parameters. Please explain.

As summarized in table 4, more parameters can provide a smaller best fit chi2 (chi2_min in the table), but due to the extra number of parameters involved, there is also larger uncertainties. Therefore as the reviewer observed, the maximum chi2 (chi2_max in the table) is larger with more parameters present.

- p18, l10-11: "The maximum uncertainties for AOD are at 410nm with a value of 0.009". How does this relate to the error in AOD of 0.017 quoted one sentence earlier? Thank you for pointing this out. The value in the manuscript is not correct. As shown in Fig 12 (a) the uncertainty at 410nm should be larger than the one at 550nm. We have updated the uncertainty at 410nm with the correct value of 0.022.

- p22, l1: "MOIDS" –> "MODIS"
Corrected in the revised manuscript.